# Unraveling the Neuropharmacological Properties of *Lippia alba:* A Scientometric Approach

**DOI:** 10.3390/ph18030420

**Published:** 2025-03-16

**Authors:** Pedro I. C. Silva, Lucas V. P. S. Pantoja, Brenda C. Conceição, Marta E. O. Barbosa, Luiza F. R. Soares, Rui Daniel Prediger, Enéas A. Fontes-Júnior, Jofre J. S. Freitas, Cristiane S. F. Maia

**Affiliations:** 1Programa de Pós-Graduação em Farmacologia e Bioquímica, Universidade Federal do Pará, Belém 66075-900, PA, Brazil; pedro.icd.silva@uepa.br (P.I.C.S.); lucas-villar@outlook.com (L.V.P.S.P.); brenda.conceicao@ics.ufpa.br (B.C.C.); 2Laboratório de Farmacologia da Inflamação e do Comportamento, Instituto de Ciências da Saúde, Universidade Federal do Pará, Belém 66075-900, PA, Brazil; marta.barbosa@ics.ufpa.br (M.E.O.B.); luiza.fernanda1232@gmail.com (L.F.R.S.); efontes@ufpa.br (E.A.F.-J.); 3Centro de Estudos Pré-Clínicos da Amazônia, Universidade do Estado do Pará, Belém 66087-662, PA, Brazil; 4Programa de Pós-Graduação em Ciências Farmacêuticas, Universidade Federal do Pará, Belém 66075-900, PA, Brazil; 5Laboratório Experimental de Doenças Neurodegenerativas, Departamento de Farmacologia, Centro de Ciências Biológicas, Universidade Federal de Santa Catarina, Florianópolis 88049-900, SC, Brazil; rui.prediger@ufsc.br

**Keywords:** *Lippia alba*, Verbenaceae, medicinal plants, ethnopharmacology, essential oil, phytochemistry, traditional medicine, phytotherapy, bibliometric analysis, neuropharmacology

## Abstract

*Lippia alba* (Verbenaceae) is popularly known as lemon balm or false melissa and is one of the most widely used plants in traditional medicine in the Amazon region. In this study, we conducted a comprehensive bibliometric analysis, with conventional metrics associated with a critical review based on the neuropharmacological activities, to identify potential medical applications and also gaps in knowledge that require further investigation. Fifty-two articles were included according to the eligibility criteria. In the country analysis, Brazil emerged as the main contributor to research with the highest number of publications and citations. Notably, nine of the ten main research institutions are Brazilian, with the Universidade Federal de Santa Maria standing out with 761 citations. The keywords “anesthesia”, “*Lippia alba*”, and “essential oil” were the most frequent, highlighting their importance in this field. Essential oils are the most common type of extraction, which linalool, citral, geraniol, carvone, and limonene were the main constituents identified. According to the type of study, preclinical studies presented the highest frequency, primarily through fish experimental models. The main neuropharmacological activities identified were sedative–anesthetic, anxiolytic, anticonvulsant, and analgesic, with mechanisms of action via the GABAergic pathway. This bibliometric review provided new evidence reinforcing the potential of *L. alba* as a promising alternative for the treatment of neuropsychiatric disorders. It also highlighted existing knowledge gaps, mainly related to the comparison of the actions of the different chemotypes of the species and the investigation of the mechanisms underlying their neuropharmacological properties. Additionally, there is a lack of knowledge in other emerging areas related to the central nervous system, such as mood and cognitive disorders.

## 1. Introduction

The use of medicinal plants as a therapeutic resource for human diseases is an ancestral practice in different cultures and civilizations [1]. Currently, approximately 80% of the population in developing countries obtains medicinal plants as the primary source of basic health care. This relationship is facilitated mainly by easy access to the plants, combined with low cost and the expectation of few adverse effects and low toxicity associated with the natural profile [2]. Thus, studies have validated the biological activities of these plants, highlighting their significance for global health by serving as essential sources in the search for new treatments and the identifications of bioactive compounds with potential medicinal use [3,4].

Amazon biome presents an extensive biodiversity, especially in flora, which is used by the native population to treat numerous illnesses. Therefore, it represents a valuable source of natural bioactive compounds that are underexplored socio-economically and scientifically [5]. In this scenario, the species *Lippia alba* emerges as a potential species extensively used in traditional communities. Taxonomically, *Lippia alba* belongs to the Verbenaceae family, widely distributed in the Amazon region, as well as in regions of the Americas, Asia, and Oceania [6]. It is characterized as a branched aromatic shrub, reaching heights between 0.6 and 2 m, with ascending and elongated tetragonal branches, petiolate leaves, and tiny flowers with violet corolla and schizocarp fruits [7].

In Brazil, *Lippia alba* is popularly known as lemon balm or false melissa, and it is one of the most used plants in traditional medicine [8]. The leaves and roots are used as infusions and syrup to treat hypertension, digestive problems, nausea, and colds, as well as to cure wounds and skin diseases, and to alleviate coughs and bronchitis [9]. Furthermore, various preparations of *Lippia alba* are used to achieve central nervous system (CNS) effects, including increasing strength or energy for spiritual cleansing and purification, and alleviating states of excitement [10].

In fact, preclinical studies demonstrate that the plant exhibits sedative–anesthetic actions, mainly in fish and aquatic invertebrates, by promoting a shorter anesthetic induction time and a longer anesthetic plane duration through the GABAergic pathway. A neuroprotective effect has been associated with this property, particularly through the enhancement of enzymatic antioxidant capacity and the reduction in lipid peroxidation, as well as through anti-inflammatory action and a decrease in systemic stress, with few side effects. Other properties investigated include anxiolytic, anticonvulsant, analgesic, anticholinesterase, and inhibitory activities on CNS excitability [11,12,13]. Clinical studies indicate that *L. alba* serves as an alternative for the treatment of migraines in women, by reducing the frequency of episodes and the intensity of pain. These results suggest that *L. alba* exhibits promising neuropharmacological activities, highlighting several potential interactions with CNS targets [14].

Considering that the investigation of the neuropharmacological effects of the *Lippia alba* has been performed using different models, methods, and approaches, gathering findings, connecting conclusions, and identifying gaps in the scientific knowledge are necessary [15]. In this context, bibliometric analysis emerges as a type of study using quantitative measures to analyze the visibility, impact, influence, and trends of scientific production within a specific field, providing support for decision-making based on scientific evidence [16].

This study aims to perform a comprehensive bibliometric-type analysis to investigate the scientific production related to the neuropharmacological properties of *Lippia alba*. The metric data collected will provide a global perspective on the main aspects of the studies, identifying gaps in the existing scientific literature on *Lippia alba*. These findings could lead to further investigation, enabling research opportunities and potential future directions.

## 2. Materials and Methods

We now employ a global bibliometric approach, as previously reported by our research group [17,18].

### 2.1. Source and Collection of Information

On 5 July 2024, we conducted a global search for *Lippia alba* and its neuropharmacological effects using the search strategy (TS = “*Lippia alba*”) in the Web of Science Core Collection (WoS-CC) database.

### 2.2. Study Inclusion Criteria and Article Selection

We included original and review articles investigating the neuropharmacological effects of *Lippia alba*, with no language restrictions. Although review studies did not present original data, such literature was included to support the critical analysis, also providing the gaps in the literature that warrant further investigation. Two independent researchers followed a structured protocol, in which each abstract was thoroughly reviewed, and if the article approached neuropharmacological effects, it was selected for inclusion. Disagreements regarding article inclusion were resolved by a senior researcher, who conducted a full-text review and critical analysis to determine the final decision. Conference papers, editorials, and articles on *Lippia alba* that did not evaluate neuropharmacological properties were excluded, as detailed in the Appendix A.

### 2.3. Bibliometric Procedures

For this study, we retrieved data from the WoS-CC database, including article titles, author names, citation counts, journal names, author keywords, countries, and institutions. Using VOSviewer software (version 1.6.16), we performed co-authorship analysis (based on publication and citation counts), evaluated keyword occurrences, and analyzed institutional contributions. The generated networks were interpreted as follows: each cluster represents an analysis item (e.g., authors, keywords, or institutions); the size of the cluster indicates the volume of publications or citations, the frequency of keyword occurrences, or institutional productivity; and the lines connecting clusters indicate co-authorship, keyword connections, or inter-institutional collaborations [17].

We also analyzed the relevance of journals, considering publication frequency and impact factors based on JCR 2023, ©2024 Clarivate. Additionally, we assessed the geographic distribution of selected articles using MapChart (https://www.mapchart.net/, accessed on 12 August 2024).

### 2.4. Critical Analysis

In addition to the bibliometric analysis, we critically reviewed the selected articles. The knowledge mapping (e.g., study design), phytochemical procedures, experimental protocols, research themes, and pharmacological approaches were evaluated. In the comprehensive analysis, we provide detailed insights into the *Lippia alba* literature and its neuropharmacological effects. Figure 1 summarizes the methodological procedures utilized.

## 3. Results

### 3.1. Bibliometric Assessment

The search strategy applied in the WoS-CC retrieved 492 articles, of which 52 studies met the inclusion criteria. The remaining excluded articles are available in Appendix A. The first article was published in 1998 and investigated the *Lippia alba* anti-inflammatory and analgesic effects in a murine model [19]. The most recent article was published in 2024, which consisted of an in silico study of the *Lippia alba* essential oil anticholinesterase activity [20]. The most cited article was published in 2010, receiving 133 citations, and investigated the *Lippia alba* anesthetic effects on fish (Table 1) [21].

**Table 1 pharmaceuticals-18-00420-t001:** Selected articles on the neuropharmacological activity of *Lippia alba* in WoS-CC.

Authors/Years	Article Title	DOI/URL	Number of Citation WoS	Article Summary
Silva et al., 2024 [20]	Molecular modeling and anticholinesterase activity of the essential oil from three chemotypes of *Lippia alba* (Mill.) NEBr. ex Britton and P. Wilson (Verbenaceae)	https://doi.org/10.1016/j.heliyon.2024.e29063	0	The authors evaluated the effects of three chemotypes of *Lippia alba* per se and associated them with the enzyme acetylcholinesterase activity, describing the mechanisms of action in vitro and in silico. The authors reported that all chemotypes tested showed inhibitory action on the activity of acetylcholinesterase enzyme.
Velasquez et al., 2023 [16]	Dendritogenic potential of the ethanol extract from *Lippia alba* leaves in rat cortical neurons	https://doi.org/10.3390/molecules28186666	0	The authors investigated the dendritogenic effects of the ethanolic extract of *Lippia alba* in cultured embryonic rat neurons. The results indicated that the administration of *L. alba* presented dendritogenic activity, mediated by the PI3K signaling pathway. The authors propose that further behavioral studies should be carried out to corroborate the therapeutic effect observed.
Finamor et al., 2023 [22]	The long-term transport of *Potamotrygon wallacei* increases lactate levels and triggers oxidative stress in its brain: The protective role of recovery and the essential oil of *Lippia alba*	https://doi.org/10.1016/j.aquaculture.2023.739461	0	The authors analyzed the anti-stress effect of *Lippia alba* essential oil in cane stingray (*Potamotrygon wallacei*) on the redox state and brain function. The authors observed that *L. alba* exerts anti-stress activity by reducing levels of oxidative stress and the inflammatory response in the animals’ brains.
Nonato et al., 2023 [23]	Antibacterial activity and anxiolytic effect in adult zebrafish of Genus *Lippia* L. species	https://doi.org/10.3390/plants12081675	2	The authors investigated the biological activities of three species of the genus *Lippia*, including anxiolytic action in a zebrafish model. Both the essential oil and the ethanolic extract of *Lippia alba* showed anxiolytic action, without altering the locomotor activity of the animals. The authors also observed that the mechanisms of action of the anxiolytic activity of the essential oil are through the GABAergic mechanisms, whereas the ethanolic extract acts through the modulation of the serotonergic system.
Becker et al., 2023 [24]	Exposure of *Hyalella bonariensis* (Crustacea, Amphipoda) to essential oils: Effects on anesthesia and swimming activity	https://doi.org/10.3390/fishes8030149	1	The authors evaluated, among other behavioral parameters, the time required for anesthetic induction and recovery of *Hyalella bonariensis* exposed to essential oils, including *Lippia alba*. The *Lippia alba* showed anesthetic activity; however, it prolonged the recovery time from anesthesia. The authors emphasize that the administration of *L. alba* did not alter the locomotor activity and behavior of the animals.
de Lima et al., 2021 [7]	Eugenol and *Lippia alba* essential oils as effective anesthetics for the Amazonian freshwater stingray *Potamotrygon wallacei* (Chondrichthyes, Potamotrygonidae)	https://doi.org/10.1007/s10695-021-01029-1	7	The authors investigated the effect of *Lippia alba* essential oil on anesthetic induction and recovery time in Amazonian freshwater stingrays (*Potamotrygon wallacei*). The authors observed that *L. alba* showed anesthetic activity with rapid recovery.
Castañeda et al., 2022 [10]	Medicinal plants used in traditional Mayan medicine for the treatment of central nervous system disorders: An overview	https://doi.org/10.1016/j.jep.2021.114746	13	In this review, the authors described the main medicinal plants used by Mayan groups to treat disorders of the central nervous system. The authors report, according to their research, that *Lippia alba* demonstrates neuropharmacological activities, such as anticonvulsant and sedative activity (through the GABAergic pathway), in addition to analgesic activity for migraines.
Rucinque et al., 2021 [11]	*Ocimum americanum* and *Lippia alba* essential oils as anesthetics for Nile tilapia: Induction, recovery of apparent unconsciousness and sensory analysis of filets	https://doi.org/10.1016/j.aquaculture.2020.735902	7	The authors evaluated the anesthetic effect of two essential oils, including *Lippia alba*, on Nile tilapia. The authors found that *L. alba* essential oil induced anesthesia in the animals.
Becker et al., 2021 [25]	Anesthetic potential of different essential oils for two shrimp species, *Farfantepenaeus paulensis* and *Litopenaeus vannamei* (Decapoda, Crustacea)	https://doi.org/10.1590/0103-8478cr20200793	8	The authors investigated the anesthetic effects of essential oils, including *Lippia alba*, on two species of shrimp. The authors found that *L. alba* essential oil decreased the sedation induction time and anesthesia in a dose-dependent manner.
Aydin and Barbas, 2020 [26]	Sedative and anesthetic properties of essential oils and their active compounds in fish: A review	https://doi.org/10.1016/j.aquaculture.2020.734999	94	The authors investigated, among other properties, the anesthetic activity of *Lippia alba* essential oil applied with different types of surfactants on fish (*Oreochromis niloticus*). The authors observed that the essential oil associated with the surfactant polysorbate 20 (T20) was the least efficient for inducing stage 2 anesthesia. The association with the surfactant polysorbate 80 (T80) was the most effective in achieving deep anesthesia. The authors also report that the duration of anesthesia did not differ between the surfactants used.
Postay et al., 2021 [27]	The effectiveness of surfactants applied with essential oil of *Lippia alba* in the anesthesia of Nile tilapia (*Oreochromis niloticus*) and their toxicity assessment for fish and mammals	https://doi.org/10.1007/s11356-020-11483-8	7	The authors reviewed the main constituents of essential oils that have sedative and hypnotic effects. Regarding *Lippia alba* essential oil, the authors demonstrate that it is used as a sedative and anesthetic for fish.
Ortega-Cuadros et al., 2020 [28]	Essential oils biological activity of the shrub *Lippia alba* (Verbenaceae)	https://doi.org/10.15517/rbt.v68i1.39153	5	The authors reviewed aspects of the use of *Lippia alba* essential oils applied in biotechnology. The authors reported that *L. alba* has sedative and anesthetic effects on several species of fish.
Hoseini et al., 2019 [29]	Application of herbal anesthetics in aquaculture	https://doi.org/10.1111/raq.12245	36	The authors reviewed the anesthetic action of the main essential oils and plant extracts used in aquaculture. The authors report that *Lippia alba* presents anesthetic activity in different species used in aquaculture. The anesthetic effect is accompanied by an anti-stress effect.
Maia et al., 2019 [30]	Hydrolate toxicity of *Lippia alba* (Mill.) N. E. Brown (Verbenaceae) in juvenile tambaqui (*Colossoma macropomum*) and its potential anesthetic properties	https://doi.org/10.1016/j.aquaculture.2018.11.058	12	The authors analyzed the anesthetic induction and recovery time of *Lippia alba* hydrolates in young tambaquis. The authors found that *L. alba* presents a dose-dependent anesthetic action.
Louchard and Araújo, 2019 [6]	Pharmacological effects of different chemotypes of *Lippia alba* (Mill.) NE Brown	https://www.blacpma.usach.cl/sites/blacpma/files/articulo_1_-_1584_-_95_-_105.pdf	0	The authors reviewed the biological and pharmacological properties of *Lippia alba* associated with its chemical composition. The authors reported that the carvone chemotype showed anxiolytic activity through the GABAergic pathway. The citral-limonene chemotype showed anxiolytic, sedative, antipyretic, and myorelaxant effects, with weak to moderate action via the GABAergic pathway in rats. Both chemotypes and the myrcene chemotype showed anticonvulsant action in mice. The authors recommend monitoring the chemotype of *L. alba* for drug development.
Souza et al., 2019 [31]	Involvement of HPI-axis in anesthesia with *Lippia alba* essential oil citral and linalool chemotypes: gene expression in the secondary responses in silver catfish	https://doi.org/10.1007/s10695-018-0548-3	16	The authors elucidated the participation of the HPI axis in the anesthetic effects of two chemotypes (citral and linalool) of *Lippia alba* essential oil in silver breams. The authors reported that the use of the linalool chemotype as an anesthetic is safe and effective for the species studied, with no disturbances to the HPI axis. On the other hand, the authors found that citral chemotype induces numerous disturbances in the HPI axis, with no recommendation as an anesthetic agent.
Almeida et al., 2019 [32]	Stress-reducing and anesthetic effects of the essential oils of *Aloysia triphylla* and *Lippia alba* on *Serrasalmus eigenmanni* (Characiformes: Serrasalmidae)	https://doi.org/10.1590/1982-0224-20190021	3	The authors investigated whether essential oils, including *Lippia alba*, can be used as anesthetics and stress-reducing agents for the transport of the species *Serrasalmus eigenmanni* Norman. The authors state that *L. alba* induced an anesthetic effect, and also affected the locomotor activity.
da Silva et al., 2019 [33]	Anesthetic potential of the essential oils of *Lippia alba* and *Lippia origanoides* in Tambaqui juveniles	https://doi.org/10.1590/0103-8478cr20181059	17	The authors evaluated the anesthetic effects of two chemotypes of *Lippia alba* essential oil in juveniles of tambaqui. The authors reported that the increase in the concentration of the essential oil resulted in a proportional decrease in the time required for sedation, deep anesthesia, and recovery.
Batista et al., 2018 [34]	*Lippia alba* essential oil as anesthetic for tambaqui	https://doi.org/10.1016/j.aquaculture.2018.06.040	26	The authors analyzed the anesthetic property of *Lippia alba* through the anesthetic induction time and the physiological response to stress in tambaqui (*Colossoma macropomum*). The authors reported that *L. alba* showed anesthetic activity, with a decrease in physiological stress at higher concentrations.
Souza et al., 2018 [35]	Nanoencapsulated Melaleuca alternifolia essential oil exerts anesthetic effects in the brachyuran crab using *Neohelice granulata*	https://doi.org/10.1590/0001-3765201820170930	8	The authors investigated the anesthetic efficiency of different natural products using the brachyuran crab, *Neohelice granulata*, as an experimental model. Regarding *Lippia alba* essential oil, no anesthetic effects were observed. The authors also reported that autotomy and deaths occur in almost all concentrations used.
da Silva et al., 2018 [36]	The essential oil of *Lippia alba* and its components affect *Drosophila* behavior and synaptic physiology	https://doi.org/10.1242/jeb.176909	12	The authors evaluated the sedative activity and the influence on synaptic transmission of *Lippia alba* essential oil and its constituents in Drosophila. They observed that *L. alba* presents a sedative effect proportional to the volume administered. Citral was mainly responsible for the sedative activity.
Wang and Heinbockel, 2018 [37]	Essential oils and their constituents targeting the GABAergic system and sodium channels as treatment of neurological diseases	https://doi.org/10.3390/molecules23051061	59	The authors reviewed the beneficial effects of essential oils and their constituents for central nervous system disorders, targeting action on the GABAergic system and voltage-gated sodium channels. The authors reported that *Lippia alba* exerts central anesthetic action through the involvement of the GABAergic system, as well as anxiolytic activity in mice.
Becker et al., 2018 [38]	Ventilatory frequency and anesthetic efficacy in silver catfish, *Rhamdia quelen*: a comparative approach between different essential oils	https://doi.org/10.1590/rbz4720170185	12	The authors evaluated the anesthetic efficacy of *Lippia alba* essential oil in fish. The authors reported that *L. alba* decreased the induction time of sedation and anesthesia.
Almeida et al., 2018 [39]	Essential oils and eugenol as anesthetics for *Serrasalmus rhombeus*	https://doi.org/10.20950/1678-2305.2018.195	8	The authors analyzed the anesthetic efficacy and swimming behavior of *Serrasalmus rhombeus* treated with essential oils, including *Lippia alba*. The authors described that the *Lippia alba* essential oil presents sedative and anesthetic effects, altering locomotor activity with no interference with the animals’ motor balance.
Bandeira et al., 2018 [40]	*Lippia alba* and *Aloysia triphylla* essential oils are anxiolytic without inducing aversiveness in fish	https://doi.org/10.1016/j.aquaculture.2017.09.023	19	The authors investigated the anxiolytic activity of two essential oils, including *Lippia alba*, in two fish species to consolidate their anesthetic effect. The authors reported that *L. alba* presents an anesthetic effect and anxiolytic activity, not altering locomotion, with good tolerance.
Salbego et al., 2017 [41]	Biochemical parameters of silver catfish (*Rhamdia quelen*) after transport with eugenol or essential oil of *Lippia alba* added to the water	https://doi.org/10.1590/1519-6984.16515	27	The authors evaluated the sedative effects of botanic compounds, including *Lippia alba* essential oil, on metabolic responses and brain acetylcholinesterase activity in fish. The authors observed that *Lippia alba* exerts anesthetic activity with a decrease in acetylcholinesterase enzyme activity in the brain.
Tsuchiya, 2017 [8]	Anesthetic agents of plant origin: A review of phytochemicals with anesthetic activity	https://doi.org/10.3390/molecules22081369	63	The authors reviewed different classes of phytochemicals with anesthetic, analgesic, or sedative properties. The authors report that *Lippia alba* reduced the excitability of the sciatic dros in rats and presented sedative and anesthetic activity in fish through interaction with the GABAergic system.
Salbego et al., 2017 [42]	Anesthesia and sedation of map treefrog (*Hypsiboas geographicus*) tadpoles with essential oils	https://doi.org/10.1590/0103-8478cr20160909	2	The authors evaluated the sedative and anesthetic activities of *Lippia alba* essential oil (citral and linalool chemotypes) in *Hypsiboas geographicus* tadpoles. The authors reported that the plant’s essential oil, for both chemotypes, had sedative and anesthetic actions.
Simoes et al., 2017 [43]	Essential oil of *Lippia alba* as a sedative and anesthetic for the sea urchin *Echinometra lucunter* (Linnaeus, 1758)	https://doi.org/10.1080/10236244.2017.1362317	5	The authors explored the sedative and anesthetic effects of *Lippia alba* essential oil on sea urchins (*Echinometra lucunter*). The authors describe that the oil presents sedative and anesthetic action on sea urchins.
Souza et al., 2017 [44]	Physiological responses of *Rhamdia quelen* (Siluriformes: Heptapteridae) to anesthesia with essential oils from two different chemotypes of *Lippia alba*	https://doi.org/10.1590/1982-0224-20160083	35	The authors investigated the sedative and anesthetic actions of two chemotypes (citral and linalool) of *Lippia alba* in jundiás (*Rhamdia quelen*). The authors also reported that the citral chemotype induced sedation more quickly than the linalool chemotype. All chemotypes induced anesthesia.
Sena et al., 2016 [45]	Essential oil from *Lippia alba* has anaesthetic activity and is effective in reducing handling and transport stress in tambacu (*Piaractus mesopotamicus x Colossoma macropomum*)	https://doi.org/10.1016/j.aquaculture.2016.09.033	41	This study analyzed the anesthetic efficacy of *Lippia alba* essential oil in juvenile tambacu (*Piaractus mesopotamicus* × *Colossoma macropomum*). The essential oil induced sedative and anesthetic effects on the species studied. The authors highlight the promising use of *L. alba* as an anesthetic for fish.
Cárdenas et al., 2016 [46]	Effects of clove oil, essential oil of *Lippia alba*, and 2-phe anesthesia on juvenile meager, *Argyrosomus regius* (Asso, 1801)	https://doi.org/10.1111/jai.13048	24	The authors determined the anesthetic efficacy of *Lippia alba* essential oil on juvenile meager, *Argyrosomus regius* (Asso, 1801). The authors reported that the essential oil presents profound anesthetic properties, with rapid recovery, related to an increase in cortisol levels, and not affecting the expression of pituitary growth hormone.
Hohlenwerger et al., 2016 [47]	Could the essential oil of *Lippia alba* provide a readily available and cost-effective anesthetic for Nile tilapia (*Oreochromis niloticus*)?	https://doi.org/10.1080/10236244.2015.1123869	38	The authors explored the ideal concentration of *Lippia alba* essential oil for anesthetic induction and recovery in Nile tilapia. The authors found that *Lippia alba* has a sedative and anesthetic action, with a concentration of 500 μL L^−1^ promising for this action in Nile tilapia.
Toni et al., 2015 [48]	Sedative effect of 2-phenoxyethanol and essential oil of *Lippia alba* on stress response in gilthead sea bream (*Sparus aurata*)	https://doi.org/10.1016/j.rvsc.2015.09.006	43	The authors evaluated the anesthetic efficacy of *Lippia alba* essential oil on the hypothalamic and pituitary hormonal expression of sea bream (*Sparus aurata*). *L. alba* presented a sedative and anesthetic effect, not affecting the response to physiological stress.
Souza et al., 2015 [49]	*Rhamdia quelen* (Quoy & Gaimard, 1824), submitted to a stressful condition: Effect of the dietary addition of the essential oil of *Lippia alba* on metabolism, osmoregulation, and endocrinology	https://doi.org/10.1590/1982-0224-20140153	23	The authors analyzed the effects of the dietary addition of *Lippia alba* essential oil in *Rhamdia quelen* on metabolic, osmoregulatory, and endocrine parameters. The authors reported that the plant does not present a protective action against stress in the evaluated species.
de Sousa et al., 2015 [15]	A systematic review of the anxiolytic-like effects of essential oils in animal models	https://doi.org/10.3390/molecules201018620	86	The authors performed a systematic review to compile information on the anxiolytic effect of essential oils in animal models. The authors reported that *Lippia alba* essential oil exerts anxiolytic effects on rodents and fish, not impairing motor activity. In fish, the anxiolytic effect was associated with the GABAergic system.
Sousa et al., 2015 [13]	Essential oil of *Lippia alba* and its main constituent citral block the excitability of rat sciatic nerves	https://doi.org/10.1590/1414-431X20154710	25	The authors investigated the actions of *Lippia alba* essential oil on compound action potentials (CAPs) in the sciatic nerve of Wistar rats. The authors found that the essential oil inhibited the excitability and conductivity of all types of myelinated fibers in the animal sciatic nerve.
Heldwein et al., 2014 [50]	S-(+)-Linalool from *Lippia alba*: sedative and anesthetic for silver catfish (*Rhamdia quelen*)	https://doi.org/10.1111/vaa.12146	60	The authors evaluated the anesthetic effect and the possible involvement of the GABAergic system of *Lippia alba* essential oil and its isolated compound, linalool, in silver catfish (*Rhamdia quelen*). The authors reported that *Lippia alba* essential oil presents a sedative and anesthetic effect that differs from its isolated compound linalool. While *L. alba* mediates an anesthetic effect through benzodiazepine mechanisms, linalool does not exert this effect.
Salbego et al., 2014 [51]	The essential oil from *Lippia alba* induces biochemical stress in the silver catfish (*Rhamdia quelen*) after transportation	https://doi.org/10.1590/1982-0224-20130178	33	The authors investigated whether the effect of rapid sedation with *Lippia alba* essential oil interferes with enzymes related to purinergic signaling throughout the brain of silver catfish (*Rhamdia quelen*). The authors found that *L. alba* essential oil presented sedative activity, not reducing the purinergic mediators’ basal levels.
Carmona et al., 2013 [3]	*Lippia alba* (Mill.) N. E. Brown hydroethanolic extract of the leaves is effective in the treatment of migraine in women	https://doi.org/10.1016/j.phymed.2013.03.017	13	The authors evaluated the therapeutic action of the hydroethanolic extract of *Lippia alba* in women with migraines. The authors reported that *L. alba* extract elicited a reduction in headache episodes and symptoms in women with migraines. Furthermore, the authors stated that there were no side effects.
Heldwein et al., 2012 [52]	Participation of the GABAergic system in the anesthetic effect of *Lippia alba* (Mill.) NE Brown essential oil	https://doi.org/10.1590/S0100-879X2012007500052	54	The authors evaluated the involvement of the GABAergic system in the anesthetic activity of *Lippia alba* essential oil in silver catfish (*Rhamdia quelen*). The authors reported that the extract presented a deep sedative and anesthetic action related to the GABAergic system activation.
Parodi et al., 2012 [53]	The anesthetic efficacy of eugenol and the essential oils of *Lippia alba* and *Aloysia triphylla* in post-larvae and sub-adults of *Litopenaeus vannamei* (Crustacea, Penaeidae)	https://doi.org/10.1016/j.cbpc.2011.12.003	82	The authors exploited the essential oils’ anesthesia induction and recovery times, including *Lippia alba*, in white shrimp and post-larvae (*Litopenaeus vannamei*). The authors observed that the *Lippia alba* showed sedative and anesthetic activity in shrimp subadults and post-larvae.
Hatano et al., 2012 [4]	Anxiolytic effects of repeated treatment with an essential oil from *Lippia alba* and (R)-(-)-carvone in the elevated T-maze	https://doi.org/10.1590/S0100-879X2012007500021	37	The authors investigated the anxiolytic effect of *Lippia alba* essential oil in Wistar rats. The authors demonstrated that *L. alba* presented anxiolytic activity, which the constituent carvone may be related to this effect.
Conde et al., 2011 [14]	Chemical composition and therapeutic effects of *Lippia alba* (Mill.) N. E. Brown leaves hydro-alcoholic extract in patients with migraine	https://doi.org/10.1016/j.phymed.2011.06.016	20	The authors investigated whether the hydroalcoholic extract of *Lippia alba* reduces the intensity and frequency of headache episodes in patients with migraine. The authors reported that the plant was effective in reducing both the intensity and frequency of headache episodes in migraine patients.
da Cunha et al., 2011 [54]	Anesthetic induction and recovery of Hippocampus reidi exposed to the essential oil of *Lippia alba*	https://doi.org/10.1590/S1679-62252011000300022	50	The authors identified the anesthetic induction and recovery times of *Lippia alba* essential oil in seahorses (*Hippocampus reidi*). The authors reported that *L. alba* presents a sedative and anesthetic action in seahorses.
da Cunha et al., 2010 [21]	Essential oil of *Lippia alba*: A new anesthetic for silver catfish, *Rhamdia quelen*	https://doi.org/10.1016/j.aquaculture.2010.06.014	133	The authors explored the anesthetic activity of *Lippia alba* essential oil on *Rhamdia quelen*. The authors demonstrated that *L. alba* presents a sedative and anesthetic action.
Neto et al., 2009 [55]	The role of polar phytocomplexes on anticonvulsant effects of leaf extracts of *Lippia alba* (Mill.) NE Brown chemotypes	https://doi.org/10.1211/jpp/61.07.0013	19	The authors evaluated the anticonvulsant activity and the involvement of the GABAergic system of different chemotypes of *Lippia alba* in mice and rats. The authors reported that the linalool and citral chemotypes exerted anticonvulsant action in rodents. The citral chemotype elicited anticonvulsant activity by the modulation of GABAergic neurotransmission.
Hennebelle et al., 2008 [56]	Antioxidant and neurosedative properties of polyphenols and iridoids from *Lippia alba*	https://doi.org/10.1002/ptr.2266	51	The authors analyzed the sedative properties of polyphenols and iridoids from *Lippia alba.* The authors reported that polar compounds from *L. alba* elicited a sedative action through interaction with benzodiazepine and type GABA_A_ receptors. The authors claimed that these results alone do not fully explain the traditional pharmacological effects of *Lippia alba*.
Zétola et al., 2002 [12]	CNS activities of liquid and spray-dried extracts from *Lippia alba*—Verbenaceae (Brazilian false melissa)	https://doi.org/10.1016/S0378-8741(02)00187-3	52	The authors investigated the anticonvulsant, sedative, and myorelaxant effects of liquid and dry extracts of *Lippia alba*. The authors reported that *Lippia alba* presented a sedative and myorelaxant effect, but the anticonvulsant effect was unclear. The authors concluded that this effect is related to non-volatile compounds in the plant leaves.
Viana et al., 2000 [57]	Anticonvulsant activity of essential oils and active principles from chemotypes of *Lippia alba* (MILL.) NE BROWN	https://doi.org/10.1248/bpb.23.1314	70	The authors studied the anticonvulsant effects of three chemotypes of *Lippia alba* essential oils in mice. The authors reported that all chemotypes displayed an anticonvulsant effect, and the compounds citral, beta-myrcene, and limonene may be responsible for this effect. According to the authors, the pharmacological profile is similar to that of GABAergic drugs.
Vale et al., 1999 [58]	Behavioral effects of essential oils from *Lippia alba* (Mill.) N.E. Brown chemotypes	https://doi.org/10.1016/S0378-8741(98)00215-3	61	The authors investigated the anxiolytic, locomotor, and myorelaxant activities, as well as the rectal temperature of mice treated with three chemotypes of *Lippia alba*, clarifying their possible mechanism of action. The authors reported that the plant displayed an anxiolytic effect, not altering locomotor activity. However, solely the citral chemotype, at a high dose, showed myorelaxant action. All chemotypes decreased the rectal temperature of the animals.
Viana et al., 1998 [19]	Analgesic and antiinflammatory effects of two chemotypes of *Lippia alba*: A comparative study	https://doi.org/10.1076/phbi.36.5.347.4646	45	The authors evaluated the antinociceptive and anti-inflammatory activity of two *Lippia alba* chemotypes in mice. The authors reported that the citral chemotype of *Lippia alba* showed central analgesic action, which is not mediated by the opioid system.

Legend—PI3K: phosphatidylinositol 3-kinase; HPI: hypothalamus–pituitary–interrenal; CNS: central nervous system.

#### 3.1.1. Institutions and Authors Production 

A total of 60 institutions were involved in the publications regarding the neuropharmacological properties of *Lippia alba* (Figure 2). The most prolific institutions are predominantly located in Brazil, with nine out of the ten leading institutions (Figure 2B,C). These institutions include the Federal University of Santa Maria (UFSM), Federal University of Western Pará (UFOPA), Federal University of Ceará (UFC), University of São Paulo (USP), Federal University of Amazonas (UFAM), Federal University of Paraná (UFPR), University of Vila Velha, Ceará State University (UEC), and Federal University of Santa Catarina (UFSC) (Figure 2B,C). Notably, the Federal University of Santa Maria stands out with 27 publications and 761 citations.

Regarding researchers, a total of 233 authors have contributed to this theme (Figure 3). Concurrently, the most prolific author, B. Baldisserotto (27 publications, 761 citations), is affiliated with the Federal University of Santa Maria and leads the main publication network related to the neuropharmacological properties of *Lippia alba* (Figure 3B–D). These data indicate that all research output from the Federal University of Santa Maria is generated by Baldisserotto’s research group. Other authors from this institution have published at least four articles, cooperating with Baldisserotto’s research network (Figure 3B–D).

#### 3.1.2. Global Distribution of Production

To visualize the scientific production of *Lippia alba*, we investigated the countries of the corresponding authors. The distribution exhibited productions in nine countries on different continents (Figure 4). Brazil presents the concentration of publications (n = 41), followed by Colombia, Spain, and the United States, all with two publications. Türkiye, Japan, Iran, Guatemala, and France reached one publication per country (Figure 4A). In terms of the number of citations per country, Brazil still leads with 76.15% of total citations (n = 1191), followed by Turkey (n = 94), the United States (71), Japan (n = 63), and France (51). The remaining countries identified presented between forty and five citations (Figure 4B). Such data showed a mean value of 29.04 citations per Brazilian publication.

#### 3.1.3. Articles by Journals

According to journal analysis, only nine out of twenty-nine journals published at least two studies related to the neuropharmacological effects of *Lippia alba* (Figure 5). Among these, the *Aquaculture* journal (8 publications, 332 citations, and JCR impact factor 3.9) exhibited the highest number of published articles, followed by the journal *Neotropical Ichthyology* (5 publications, 144 citations, and JCR impact factor 2.0). Importantly, both journals focus on research involving fish physiology and pharmacology, highlighting that the majority of studies on *Lippia alba* pharmacological activity are conducted in fish species. The full list of journals is available in Appendix A.

#### 3.1.4. Keywords Presented

A co-occurrence keyword analysis reveals the logical structure of the prevailing research trends and the scientific terms that characterize the field of study. Our survey identified 174 terms associated with the selected papers (Figure 6). The top 10 most frequent keywords were anesthesia, *Lippia alba*, essential oil, stress, citral, cortisol, recovery, sedation, anxiety, and behavior (Figure 6B). It is noteworthy that these terms indeed represent the key aspects of our research.

#### 3.1.5. Scientific Production by Decades

A clear majority of publications occurred in the 2010s (n = 34), significantly more than in the 2020s (n = 12), 2000s (n = 4), and 1990s (n = 2). This decade also received the most citations (845), exceeding the 2000s (192), 2020s (144), and 1990s (106). See Figure 7 for a comprehensive overview.

### 3.2. Critical Analysis

The selected studies on the neuropharmacological properties of *Lippia alba* were categorized into pre-clinical in vivo studies, clinical studies, and literature reviews (Figure 8). Pre-clinical in vivo studies comprised 42 articles with mixed methodologies (e.g., incorporating in silico or in vitro methods). Among the experimental models, fish were most frequently used (59.52%), followed by rodents (16.66%), and crustaceans (9.52%), with in vitro and in silico methods representing one study per method. Pre-clinical studies dominated, accounting for 80.76% of the publications (Figure 8).

We employed critical content analysis to enhance science mapping, a main goal of bibliometric analysis (Appendix A; Figure 9). The first part of the mapping involved the phytochemical procedures used in the studies. We found that the leaves, flowers, and branches of *L. alba* were primarily used for extraction. The most common methods were hydrodistillation in a Clevenger-type apparatus, percolation, and maceration. Essential oils constituents were characterized in 38 studies, while methanolic, ethanolic, and hydroethanolic extract components were described in six studies. The most prevalent phytoconstituents identified were linalool, citral, geraniol, and carvone (Figure 9). Additional details, including plant parts, extraction methods, and chemical compositions are provided in Appendix A. Regarding pharmacological aspects, the most frequent neuropharmacological activities reported were sedative, anesthetic, anxiolytic, anticonvulsant, and analgesic. The doses and concentrations of essential oil or extracts of *Lippia alba* differ significantly between studies (Appendix A). Figure 8 summarizes the main neuropharmacological activities, geographic locations, major chemical components, and types of extracts identified in the selected studies.

## 4. Discussion

Phytotherapy represents a valuable strategy for managing numerous diseases [59]. In this context, scientific approaches are vital for determining the safety and efficacy of medicinal plants. Our research focused on *Lippia alba*, a relevant medicinal plant in folk medicine primarily used to treat CNS disorders. By analyzing global research output on the neuropharmacological properties of *Lippia alba*, a research gap was identified: only a few scientific investigations have explored this plant. Additionally, a scarcity of pharmacological models for this species was found, highlighting that the majority of studies were focused on aquaculture applications.

Bibliometric analysis guides academics and researchers toward a discipline’s most influential studies, mapping the field to discover trends and new topics of investigation [60]. Our bibliometric study adopted the three major laws of bibliometrics, focusing on author productivity (Lotka’s Law), journal performance (Bradford’s Law), and the most frequent keywords (Zipf’s Law). Regarding Lotka’s Law, we identified that most researchers focused on the research of the neuropharmacological properties of *Lippia alba* were from Brazil. Moreover, the most influential author is also a Brazilian researcher. Baldisserotto appears to be a prolific scientist in this field, with his scientific network being the most important identified in this study.

Considering the analysis of institutions, it is noteworthy that author productivity influences institutional performance. For example, the Federal University of Santa Maria stands out as the most productive and cited institution. Interestingly, the scientific metrics for this institution were related to studies developed by Baldisserotto’s research group, indicating that this author significantly contributed to the institution’s excellent ranking. Additionally, the main institutions publishing on this topic are also from Brazil, highlighting the positive impact of the country’s rich biodiversity. We also identified a lack of international scientific connectivity between the analyzed institutions, except for the Federal University of Santa Maria, the Federal University of Pará, and the Federal University of Western Pará, which cooperated with two Spanish institutions, the University of Cadiz and the Consejo Superior de Investigaciones Científicas (CSIC). Such findings reflect a need for increased international collaboration to enhance research impact and knowledge sharing.

The application of Bradford’s Law revealed that research on the neuropharmacological effects of *Lippia alba* is concentrated in a few specific journals, with only nine out of twenty-nine periodicals publishing multiple studies on the topic. *Aquaculture* and *Neotropical Ichthyology* were identified as the leading journals, indicating a predominant focus on fish physiology and pharmacology, indicating that the fish model represents the primary source for research on the topic, according to Bradford’s Law. This suggests that the majority of studies on *Lippia alba* are conducted within an aquatic context, emphasizing the plant’s application in aquaculture.

In line with this, the keyword analysis of research on *Lippia alba* sheds light on the main trends and focus areas within the field. The top keywords were anesthesia, *Lippia alba*, essential oil, stress, citral, cortisol, recovery, sedation, anxiety, and behavior. These findings indicate that research on the neuropharmacological effects of *Lippia alba* is predominantly focused on anti-stress and anxiolytic activities, as well as its use in sedation and anesthesia. This emphasizes the plant’s potential therapeutic applications and offers the scientific community a valuable source of natural products with neuropharmacological potential.

Regarding publications by decade, the 2010s were by far the most productive decade, with 34 publications and the highest number of citations (945). Although still ongoing, the 2020s demonstrate increasing research activity, with 12 articles published to date and a growing citation count (144). Notably, despite having fewer publications (four in the 2000s and two in the 1990s) compared to more recent decades, these earlier periods still garnered substantial citation numbers (192 and 106, respectively), highlighting the enduring relevance of early work. These metrics, in addition to the number of studies, reflect sustained interest in the topic. Another relevant parameter to analyze is the type of studies most published. Among the 52 articles selected for this review, in vivo preclinical studies were the most prevalent (n = 38), followed by reviews (n = 8), in vitro studies (n = 3), clinical studies (n = 2) and studies using combined in vitro and in silico models (n = 1).

Considering geographic criteria, global publications on *Lippia alba* and its neuropharmacological properties reveal a significant concentration of scientific output in Brazil, which holds the highest number of publications and citations. This leadership is likely attributed to the species’ origin and the region’s rich biodiversity. Following Brazil, the United States, Spain, and Colombia each contributed two publications. Despite having fewer publications, the United States and Spain achieved notable citation counts, with 71 and 40 citations, respectively, while Colombia received five citations. Other countries produced only one publication each. These findings underscore Brazil’s leading role in this area of research. However, it is important to emphasize that other countries have also made significant contributions to understanding the neuropharmacological properties of the species over the years. For instance, Iran, Turkey, Spain, Japan, and France have made key contributions to understanding its sedative–anesthetic action [8,26,29,31,56]. Additionally, Guatemala and the United States have provided excellent reviews on the plant and its biological activities [10,37]. Colombia has presented a study on the dendritogenic potential and a detailed review of *L. alba* [16,28]. These data highlight the fundamental contributions of these countries to advancing the understanding of the neuropharmacological properties of *Lippia alba*.

The scientific production of *L. alba* is concentrated in Latin America, particularly Brazil. A deeper analysis of the collection sites used in experimental studies reveals that Rio Grande do Sul is the primary location for collecting this species (n = 24), followed by Ceará (n = 6), and Pará (n = 6). Interestingly, Rio Grande do Sul not only leads in *L. alba* collection sites but also boasts the primary author, the highest number of publications, and the most citations related to the species, as detailed in previous sections.

Regarding the phytochemical approach, leaves [3,4,7,9,14,16,19,21,31,40,47,48] and aerial parts (including leaves, flowers, and fine branches) [12,20,30,36] were primarily used for essential oil and plant extract production. Hydrodistillation [34,36,38,39,40] and steam distillation [4,19,21,57,58] were the predominant techniques for extracting volatile compounds and essential oils, often employing a Clevenger-type apparatus [40,43,44,46]. Maceration [3,14,55] and percolation combined with maceration [12,16] were the most common methods for extract preparation, with ethanol [3,12,14,16,55] and methanol [9] as primary solvents.

Twenty-three of the 52 selected studies conducted phytochemical screening [13,14,20,30,34,36,52,55]. The most frequently reported phytochemical compounds include linalool [14,24,25,52,53], citral [20,27,30,34,36], geranial [20,30,32,34,36], geraniol [20,27,30,36,38,44], carvone [4,13,20,27,30,36,38,55], and limonene [27,36,38,39,52]. Thus, it was observed that the chemical composition of the essential oils of *L. alba* showed significant variation in their main constituents.

In fact, several studies have indicated that not only species of the *Lippia* genus but also various plants of the Verbenaceae family exhibit wide variation in the components of their essential oils, displaying a broad range of chemotypes for these species [61]. Several factors can influence the chemotype of the species, including the region of the plant used, the harvest time, the collection location, seasonal variations, climate, extraction method, and abiotic factors [62].

The main constituents of most chemotypes of the species *L. alba*, primarily linalool, citral, and carvone, are known to have effects on the CNS. Among these activities, the anti-inflammatory, antiproliferative, antinociceptive, analgesic, anxiolytic, antidepressant, and neuroprotective have been documented [63,64,65,66,67,68,69]. Therefore, the findings mentioned above may explain the neuropharmacological therapeutic effects of *L. alba* observed in this bibliometric study.

Experimental protocols revealed that essential oil was primarily administered to fish, crustaceans, amphibians, and echinoderms via a single-dose addition to tank water to assess sedative–anesthetic effects [48,51,54]. In rodents, intraperitoneal injection and gavage were the predominant routes of administration for essential oil, methanolic, and ethanolic extract administration to evaluate anxiolytic and anticonvulsant properties [19,55]. These findings indicate a promising action of the plant on the CNS.

*Lippia alba* exhibited a wide range of neuropharmacological properties. In vivo studies identified eight primary biological activities, with sedative and anesthetic (n = 31), anxiolytic (n = 3), anticonvulsant (n = 3), and central analgesic (n = 3) effects commonly reported [19,22,23,27,55]. Doses varied significantly across species: echinoderms (50–150 μL/L^−1^) [43], insects (0.5–1 μL) [36], crustaceans (250–8000 μL/L^−1^) [35], fish (5–500 μL/L^−1^) [26,30,31,33], and rodents (0.6–400 mg/kg) [12,19,57,58]. Human dosages ranged from 1 to 1.5 drops/kg/day. In vitro concentrations tested ranged from 0.01 to 230.4 mol/L [16,20].

The sedative–anesthetic activity was the most prevalent among the identified neuropharmacological properties of *Lippia alba*. Fish were the primary subjects used in the reviewed studies. Key findings include shortened sedation and anesthesia induction times coupled with prolonged anesthetic recovery, with these effects correlating to increased concentrations [30,38]. The GABAergic pathway was identified as the primary mechanism of action. Additional mechanisms involved acetylcholinesterase (AChE) inhibition and reduced calcium influx [31,36,41].

In addition to studying the sedative and anesthetic properties of *L. alba*, researchers observed the effects of the species on stress parameters in fish. It is known that stress responses in fish are controlled by the hypothalamic–pituitary–interrenal (HPI) axis. This response begins in the hypothalamus, which receives signals from the peripheral and central nervous systems. Stress signals stimulate the hypothalamus, particularly in the preoptic area, leading to the release of corticotropin-releasing hormone (CRH). Consequently, the pituitary gland responds to CRH by releasing adrenocorticotropic hormone (ACTH) into the bloodstream, which then reaches the fish’s kidneys. This hormone, in turn, induces the release of cortisol [70,71].

Cortisol, upon passing through the plasma membrane, binds to the glucocorticoid receptor (GR). This hormone-receptor complex translocates to the cell nucleus, where it activates the transcription of effector genes [72]. For this hormone-receptor binding to occur, two heat shock proteins (Hsps), Hsp70 and Hsp90, are essential. Another important protein for the modulation of the HPI axis is proopiomelanocortin (POMC), which is related to stress adaptation and environmental adjustment [73,74].

In fish treated with *L. alba*, an inhibition of the HPI axis was observed, evidenced by a decrease in the levels of CRH, cortisol, Hsp70, and Hsp90, along with an upregulation of POMCa [75,76]. These findings suggest the potential use of *L. alba* as a phytotherapeutic agent for managing stress-related disorders, such as depression [77]. However, little is known about the sedative and anesthetic effects of the species in other experimental models, such as rodents.

Regarding the side effects of the sedative–anesthetic properties of *L. alba*, it is noteworthy that most studies have not reported changes in locomotor activity or cholinergic neurotransmission. However, toxicological effects were noted in a species of crab, which exhibited mortality of all subjects, but not sedative or anesthetic response [35]. Further studies are needed to assess the safety of *L. alba* as a sedative–anesthetic in different species.

Oxidative stress and neuroinflammation play central roles in the development and progression of neuropsychiatric disorders, including depression, schizophrenia, bipolar disorder, Alzheimer’s, and Parkinson’s [78,79,80,81]. This is primarily due to the brain’s high oxygen consumption, which induces vulnerability to oxidative stress compared to other organs [82].

Oxidative stress is characterized by an imbalance between the production of oxygen (ROS) and nitrogen (RNS) free radicals and a reduced antioxidant capacity, leading to cellular damage to lipids, proteins, and DNA [82]. Alongside this process, neuroinflammation—characterized by the activation of glial cells (microglia and astrocytes) and the release of pro-inflammatory cytokines—contributes to synaptic dysfunction, neuronal death, and alterations in brain plasticity [83]. These processes negatively affect mood, cognition, and behavior, while also limiting the effectiveness of conventional therapies.

In this context, it was investigated whether *L. alba* could influence neuroinflammation and oxidative stress. In fact, fish treated with *L. alba* essential oil showed an increase in the levels of key antioxidant enzymes (SOD, CAT, GPx, GSH, and Gr) and a decrease in lipid peroxidation in their brains, indicating a neuroprotective effect [22]. Additionally, there was inhibition of inflammatory pathways, as evidenced by the downregulation of nuclear factor kappa B (NF-κB) [22]

It is important to emphasize that oxidative stress can activate the NF-κB pathway. NF-κB, in turn, can induce NADPH oxidase 2, an enzyme responsible for the production of O_2_^•−^, creating a positive feedback loop in which the activation of NF-κB by oxidative stress leads to the increased production of free radicals [84,85]. This evidence suggests that *L. alba* may serve as a promising antioxidant and anti-inflammatory agent for the CNS.

The anxiolytic effect was evaluated primarily in fish and rodents. In fish, an increase in the distance traveled, average speed, number of crossed lines, and the number of entries into the upper zone of the apparatus were observed. These effects were attributed to GABAergic and serotonergic mechanisms. No side effects on locomotor activity or cortisol levels were noted at low doses [4,23,40]. In rodents, an anxiolytic effect was also observed; however, the mechanisms of action were not investigated. This scenario suggests a broad area for future research on the anxiolytic and antidepressant effects of *L. alba* in rodents.

Unlike its anxiolytic properties, the anticonvulsant activity of *L. alba* has only been investigated in rodents (rats and mice). In these studies, a decrease in the number and intensity of seizures was observed, along with a reduction in animal mortality. Few side effects were noted, particularly with regard to locomotor activity. The mechanisms of action of this activity have been minimally explored; however, GABAergic neurotransmission also appears to underly the anticonvulsant properties of the plant [12,55,57].

The main chemical compounds identified in studies of anticonvulsant activity were citral and linalool. These compounds have demonstrated sedative and anticonvulsant effects through the involvement of the GABAergic system and the antagonism of the glutamatergic system, respectively [86,87]. These data may help explain the action of *L. alba* as an anticonvulsant and sedative, suggesting its potential as an adjuvant in the treatment of seizures in patients with epilepsy.

Changes in dendritic arborization, such as irregularities in the formation of dendritic spines and branches, are linked to the pathophysiology of neuropsychiatric disorders, including Alzheimer’s disease, autism spectrum disorder, schizophrenia, and cognitive and mood disorders, such as depression and anxiety disorders [88,89,90,91,92]. In this context, *L. alba* emerges as a potential alternative in the treatment of these pathologies due to its dendritogenic property. Velasquez and collaborators (2023) [16] demonstrated that the ethanolic extract of *L. alba* increases the total length, complexity, and dendritic branching of cortical neurons in rats. The authors observed that the dendritogenic action was mediated by the phosphatidylinositol 3-kinase (PI3K) pathway, which plays a crucial role in neuronal survival and dendrite growth.

Regarding preclinical studies, only one study investigated the mechanism of analgesic activity of *L. alba*, excluding the involvement of the opioidergic system [19]. *L. alba* also demonstrated inhibitory properties on nervous excitability and anticholinesterase activity [13,16,22]. Taken together, these findings indicate a broad spectrum of neuropharmacological actions for *L. alba*, with potential for significant therapeutic advances. Notably, the effects of the plant on other CNS disorders, such as depression and neurodegenerative diseases, remain unexplored, presenting a promising avenue for future research.

Regarding clinical studies, only two studies were identified that investigated the effects of *L. alba* on migraine in humans. Migraine is a neurovascular disorder characterized by headache and hypersensitivity to normal afferent stimuli, including light, sound, and head movements, often associated with autonomic symptoms. Its pathophysiological process is poorly understood [93,94]. Current treatments include beta-blockers, anticonvulsants, antidepressants, and others; however, their clinical efficacy is low [95].

In this context, an alternative treatment with the hydroalcoholic extract of *L. alba* rich in geranial–carvonene was proposed, with 1 to 1.5 drops per kg of body weight per day (drops/kg/day) for 2 months, in humans clinically diagnosed with migraine. A reduction in the frequency and intensity of headaches was observed, with an average clinical response time of 11 days. The authors report that there were no side effects. However, 8.3% of the treated patients experienced worsening of their clinical condition, demonstrating the heterogeneity of the disease [3,14]. This evidence points to the possibility of developing natural medications or supplements.

Based on all this evidence, *Lippia alba* stands out among other plants of the *Lippia* genus in the *Verbenaceae* family for presenting the greatest range of studied neuropharmacological activities and mechanisms of action. Studies involving species such as *Lippia multiflora*, *Lippia gracilis*, *Lippia grata*, *Lippia origanoides*, *Lippia graveolens*, *Lippia geminado*, and *Lippia adoensis* demonstrate that these species exhibit prominent analgesic activity, with few studies investigating other neuropharmacological activities for this genus of plants [96].

Chemical composition is essential in medicinal plant research, as it validates pharmacological findings and helps elucidate underlying mechanisms of action [6,28,29]. Notably, a significant proportion of the reviewed studies lacked detailed phytochemical analyses. Studies reporting the phytochemical composition revealed varied primary constituents of *L. alba* in different neuropharmacological activities. For example, linalool and citral predominated in sedative–anesthetic studies, while linalool, carvone, citral, and limonene were more common in anxiolytic and anticonvulsant studies. Analgesic studies frequently identified citral, geraniol, and carvone. Furthermore, pharmacological effects may arise from a synergistic combination of compounds, termed a phytocomplex, rather than from a single constituent [19,32,44,57].

Despite preclinical and clinical evidence supporting the neuropharmacological properties of *L. alba* and its primary constituents, no corresponding drug or herbal agent has been developed to date. This gap is common in medicinal plant research, particularly in developing regions such as South America, where most studies related to the CNS of *L. alba* are conducted. This review emphasizes the need for further research on the neuropharmacological potential of *L. alba*, guiding future investigations through knowledge gaps that include comparing the neuropharmacological effects of the various chemotypes of *L. alba*; deepening the investigation of the mechanisms of action of its activities; verifying the effects found in fish and invertebrates and studying them in other animal models.

In addition, this bibliometric review showed that *L. alba* presents promising antioxidant, anti-inflammatory, and neuroprotective activity and can be used as a therapeutic in neurological disorders associated with neuroinflammation and oxidative stress. However, most of the discoveries were generated from experimental models, and there is a lack of randomized clinical studies in humans to validate their efficacy, determine safe doses, and investigate possible drug interactions. Thus, further studies are needed to follow these future directions.

Bibliometric analysis is based on significant metric elements, allowing for the identification of emerging topics and dynamic, current trends on a given subject. It is a purely quantitative evaluation study, which constitutes one of its main limitations. Another limitation of this review is the lack of analysis regarding the methodological quality and scientific evidence of the selected articles. Consequently, the results presented are not sufficient to support public or collective health decisions; however, they highlight the main gaps in studies on the mechanisms of action of the neuropharmacological effects of *L. alba*, emphasizing the importance of researchers recognizing the need for clinical studies involving the species. The mapping conducted in this study encourages the development of new primary and secondary research on the use of *L. alba.*

## 5. Conclusions

This unprecedented study represents a significant advance in the understanding of the neuropharmacological effects of *Lippia alba*, a plant widely distributed in the Americas, especially in the Amazon biome, and extensively used in traditional medicine. This species demonstrates the ability to interact with and modulate specific mechanisms on the CNS, exhibiting sedative–anesthetic, anxiolytic, anticonvulsant, and analgesic activity, especially through the GABAergic pathway as the main mechanism of action. However, there is a significant discrepancy between the increased interest in investigating the therapeutic potential of this medicinal plant and the limitations observed in scientific research, particularly regarding its pharmacological mechanisms of action, explored primarily in the aquaculture field. Additionally, there are no studies investigating the effects of this species on cognition and other aspects of emotionality (e.g., depression) or the potential as a therapeutic agent for neurodegenerative diseases, and other brain disorders. In this context, this bibliometric analysis provides a broad overview of existing studies on the neuropharmacological activities of *L. alba*, highlighting the need for more research and greater collaboration between research groups to address these gaps and advance the neuropharmacological therapeutic potential of this promising plant, which could lead to the development of new drugs or phytotherapeutics.

## Figures and Tables

**Figure 1 pharmaceuticals-18-00420-f001:**
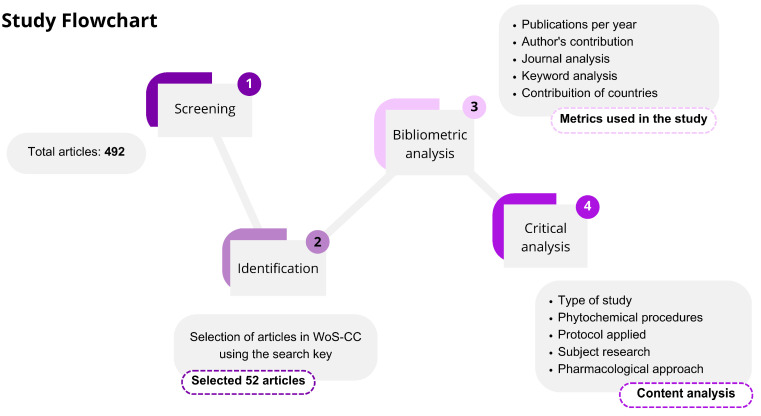
Methodological strategy applied to perform critical analysis.

**Figure 2 pharmaceuticals-18-00420-f002:**
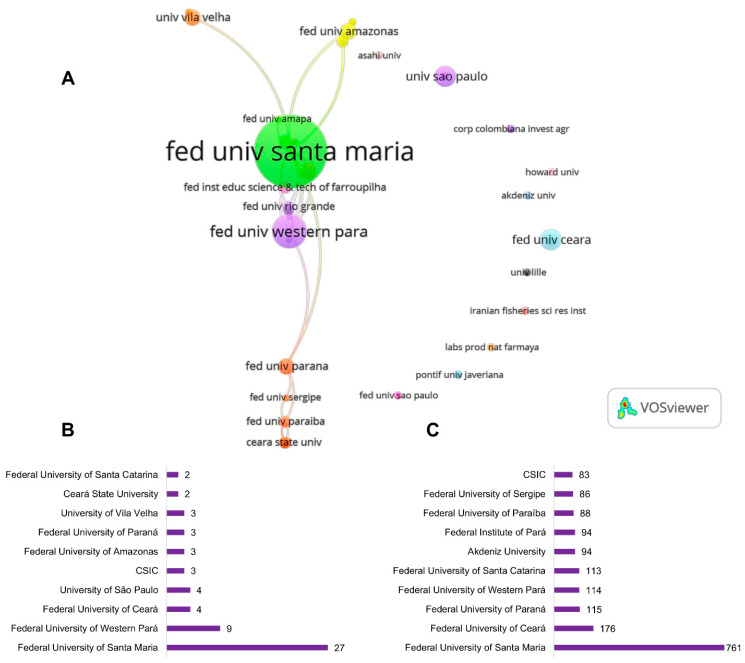
Contributions of institutions (**A**) the top 10 most productive institutions (**B**), and the most cited institutions (**C**) concerning the available literature of *Lippia alba* neuropharmacological research.

**Figure 3 pharmaceuticals-18-00420-f003:**
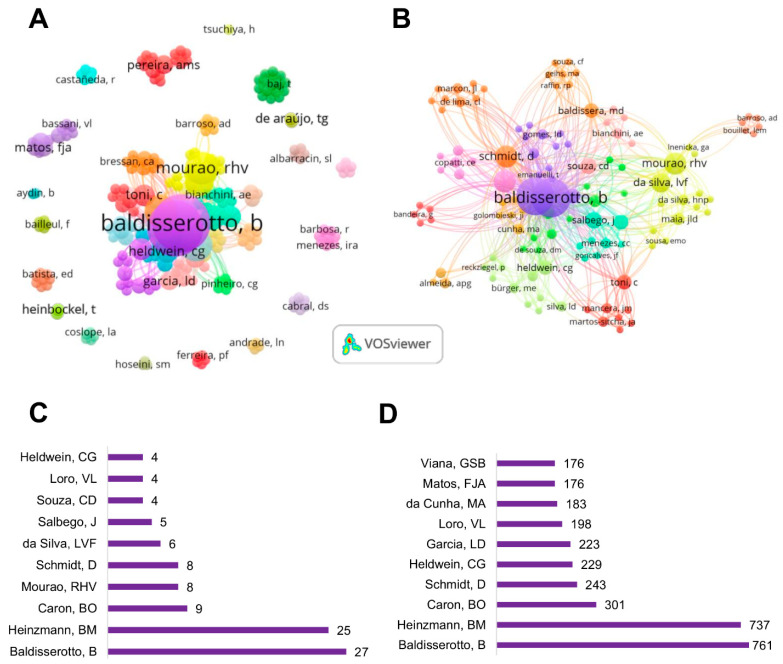
Global network visualization of authors (**A**) leading author network based on publication count (**B**), productivity (**C**), and citation impact; (**D**) *Lippia alba* neuropharmacological research.

**Figure 4 pharmaceuticals-18-00420-f004:**
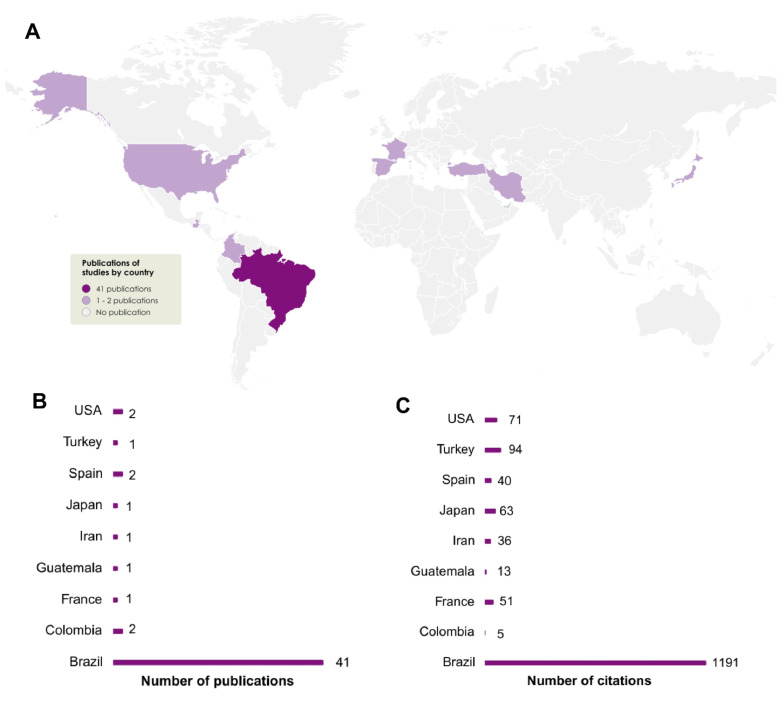
Representation of the global distribution of selected publications (**A**) description of the publication by country and (**B**) citation number by country; (**C**) *Lippia alba* neuropharmacological research.

**Figure 5 pharmaceuticals-18-00420-f005:**
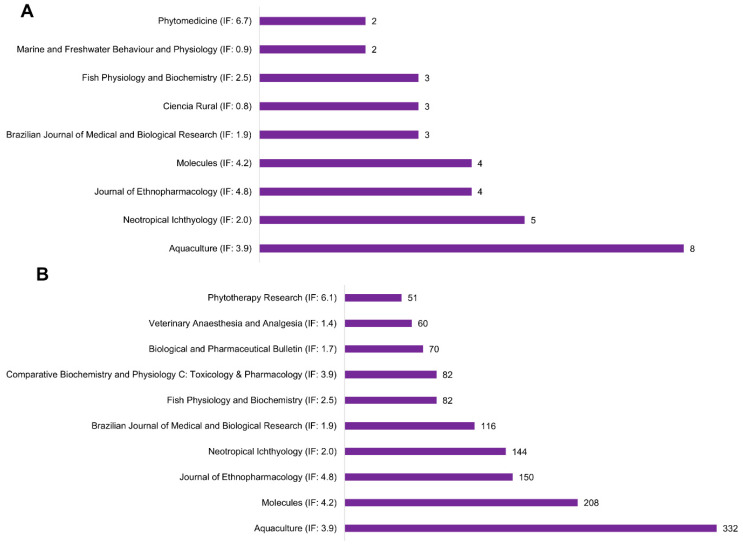
Journals with a minimum of two publications (**A**) and the top 10 most cited journals (**B**) about *Lippia alba* neuropharmacological research.

**Figure 6 pharmaceuticals-18-00420-f006:**
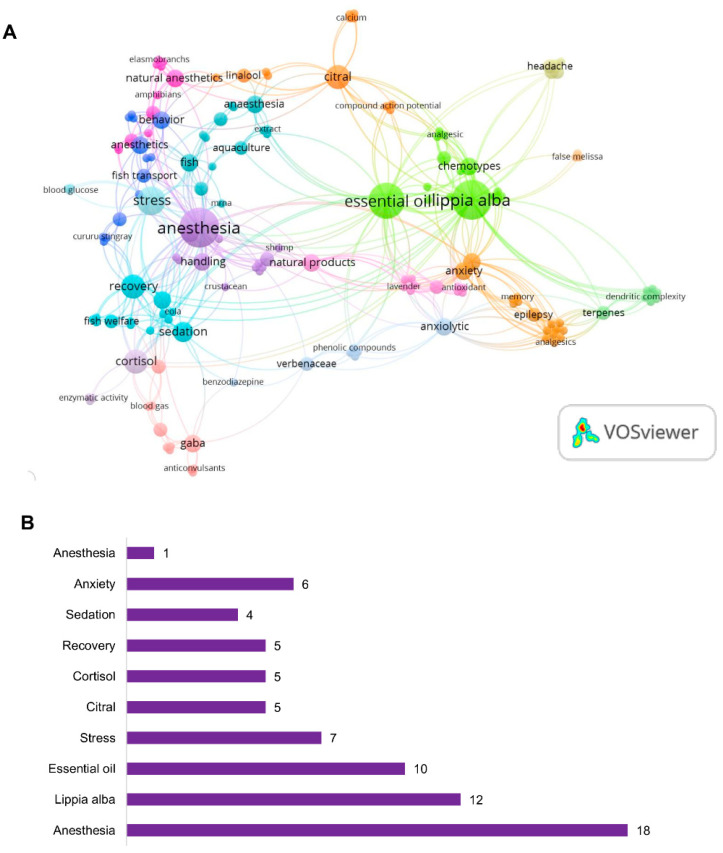
Network visualization of co-occurring keywords (**A**) with the top 10 most frequent keywords (**B**) of *Lippia alba* neuropharmacological research.

**Figure 7 pharmaceuticals-18-00420-f007:**
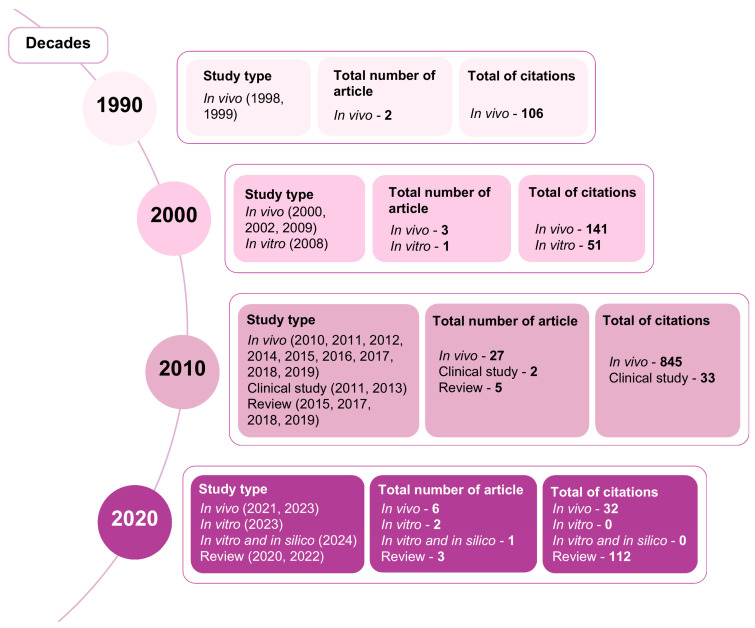
Decade-based publication, study type, and citation analysis.

**Figure 8 pharmaceuticals-18-00420-f008:**
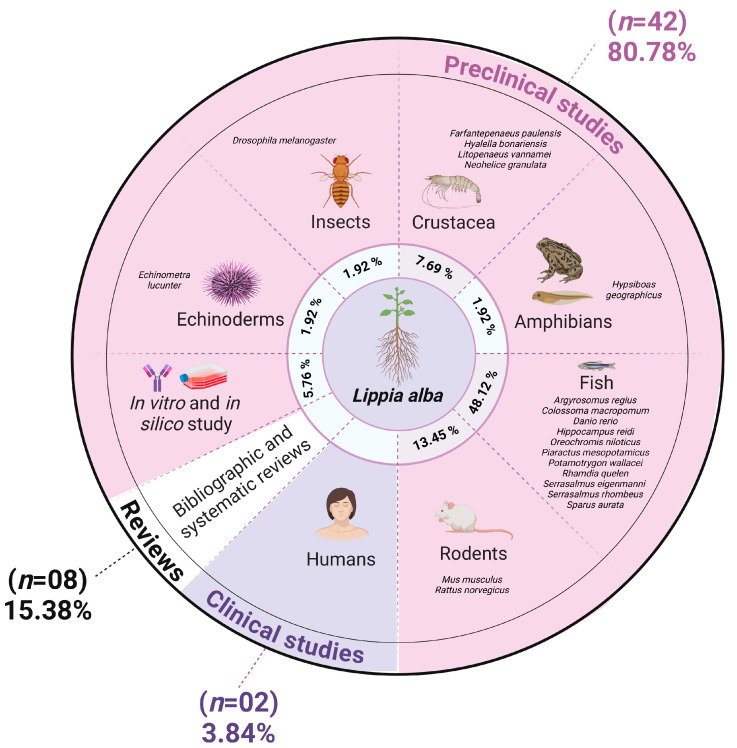
Types of studies and animal models utilized for the research of *Lippia alba* neuropharmacological effects.

**Figure 9 pharmaceuticals-18-00420-f009:**
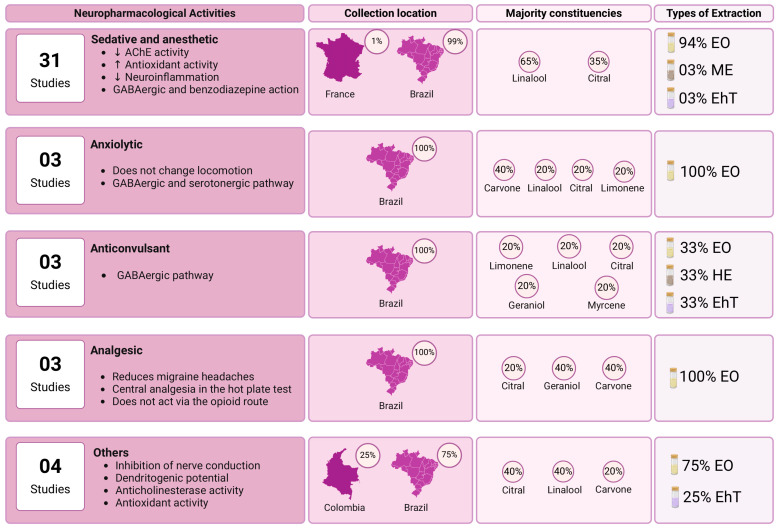
Schematic science mapping of neuropharmacological activities, geographic distribution, key chemical components, and extract types identified in reviewed studies. Legend: EO = essential oil; HE = hydroethanolic; ME = methanolic; EhT = ethanolic; AChE = Anticholinesterase enzyme.

## Data Availability

Not applicable.

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
