# Peer review of "Unraveling the Neuropharmacological Properties of Lippia alba: A Scientometric Approach"

_pharmaceuticals, 2025, doi:10.3390/ph18030420_

Round 1

Reviewer 1 Report

Comments and Suggestions for Authors

The present manuscript reports the bibliometric analysis of L. alba in protecting neurological diseases. The present form of the manuscript can be improved in the following ways.

·       Figure 1: The text under point 2 (total articles, articles read, and deleted files) should be at point 1 (selected 52 articles) and vice versa. Please look into it.

·       The reasons for including the review articles are not justified. What novel information do these articles provide?

·       Authors have included L. alba research articles and review articles on neurological diseases from WOS-CC, please explain the reasons for not selecting articles from PubMed and Scopus. There is a possibility of L. alba-mediated neurological activities.

·       Though the authors have aimed to identify the gaps for L. alba in combating neurological studies, but authors have not reported such kind of information. For example, several phytochemicals are present in this plant, please discuss, are these phytochemicals are present only in this plant? Report whether these phytochemicals have neuroprotection actions? Discuss the limitations of L. alba and elaborate on how those limitations can be overcome? Any potential side effects of L. alba? Authors must address these issues. Authors can refer to doi: 10.1002/ptr.8122.

·       The novelty of this study must be emphasized in the abstract as well as conclusions.

·       The neuroprotective mechanisms of L. alba must be highlighted in the entire manuscript.

Author Response

Dear Editor,

We first wish to thank you for the opportunity to allow a resubmission of a revised and now ameliorated manuscript. We extend our thanks to the Reviewers for their positive evaluation of our study and for their generous suggestions to clarify the manuscript: their time and efforts were very warmly appreciated. We addressed all their criticisms and, guided by the suggestions, we revised the manuscript accordingly. The detailed answers to each of the questions raised are tendered point-by-point, in order of appearance, as follows:

Reviewer #1

The present manuscript reports the bibliometric analysis of L. alba in protecting neurological diseases. The present form of the manuscript can be improved in the following ways.

  1. Figure 1: The text under point 2 (total articles, articles read, and deleted files) should be at point 1 (selected 52 articles) and vice versa. Please look into it.

Response: Thanks for this meticulous observation in Figure 1. We have kindly modified it according to your suggestions.

  1. The reasons for including the review articles are not justified. What novel information do these articles provide?

Response: Thank you for your valuable question. The inclusion of review articles in our research was a deliberate choice to provide peculiar insights, particularly given the scarcity of original studies on the neuropharmacological properties of Lippia alba. While we acknowledge that review articles do not present new data, they are essential for synthesizing existing knowledge. In this study, review literature played a pivotal role in supporting our critical analysis and highlighting significant gaps in the literature that warrant further investigation. This approach not only contextualizes our work within the field but also underscores areas requiring future research. To clarify such relevant issue, we added to the Material and Methods the sentence: “Albeit review studies did not present original data, such literature were included to support the critical analysis, also providing the gaps in the literature that warrant further investigation.” (lines: 105-107).

  1. Authors have included L. alba research articles and review articles on neurological diseases from WOS-CC, please explain the reasons for not selecting articles from PubMed and Scopus. There is a possibility of L. alba-mediated neurological activities.

Response: We appreciate the opportunity to clarify this matter. In Health science, the majority and most robust journals are indexed in the WOS-CC, which supports a relevant source of data. In addition, bibliometric analysis is a type of bibliographic study that focuses on evaluating metrics, such as the most cited authors, the journals that have published the most on a given topic, and the most frequently used keywords in the field. In this kind of study, we can select the most comprehensive database, according to the theme of the scientometric study. Thus, the WoSCC was chosen.

Our research group has published bibliometric analyses using the WoSCC database, which provides a comprehensive parameter for this type of study [1, 2].

  1. Maia, M.L.F.; Pantoja, L.V.P.S.; Da Conceição, B.C.; Machado-Ferraro, K.M.; Gonçalves, J.K.M.; Dos Santos-Filho, P.M.; Lima, R.R.; Fontes-Junior, E.A.; Maia, C.S.F. Ketamine Clinical Use on the Pediatric Critically Ill Infant: A Global Bibliometric and Critical Review of Literature.  Clin. Med.202312, 4643. https://doi.org/10.3390/jcm12144643
  2. Conceição, B.C.d.; Silva, T.A.d.; Pantoja, L.V.P.d.S.; Luz, D.A.d.; Cardoso, E.K.S.; Reis, L.D.d.S.; Raiol-da-Silva, M.C.; Kussler, M.S.; Maia, C.S.F.; Fontes-Júnior, E.A. Amazonian Plants: A Global Bibliometric Approach to Petiveria alliacea Pharmacological and Toxicological Properties. Plants202312, 3343. https://doi.org/10.3390/plants12183343

  1. Though the authors have aimed to identify the gaps for L. alba in combating neurological studies, but authors have not reported such kind of information. For example, several phytochemicals are present in this plant, please discuss, are these phytochemicals present only in this plant? Report whether these phytochemicals have neuroprotection actions? Discuss the limitations of L. alba and elaborate on how those limitations can be overcome? Any potential side effects of L. alba? Authors must address these issues. Authors can refer to doi: 10.1002/ptr.8122.

Response: We appreciate your valuable questions. We inform that the gaps regarding neurological disorders have been addressed in the manuscript concerning: i) the chemical composition of the species L. alba and the vast majority of species in the Lippia genus, as well as the Verbenaceae family, which show great variability among the studies analyzed. Thus, the authors discussed this phytochemical variation and its pharmacological implications, highlighting L. alba as a promising agent for neuropsychiatric disorders, as stated in the discussion section: “In fact, several studies have indicated that not only species of the Lippia genus but also various plants of the Verbenaceae family exhibit wide variation in the components of their essential oils, displaying a broad range of chemotypes for these species [61]. Several factors can influence the chemotype of the species, including the region of the plant used, the harvest time, the collection location, seasonal variations, climate, extraction method, and abiotic factors” (lines 344-349).

It is important to emphasize that the main chemical compounds found in the species (linalool, citral, and carvone) play a significant role in the central nervous system, particularly in neuroprotection through the attenuation of oxidative stress and neuroinflammation, as stated in the sentence: “The main constituents of most chemotypes of the species L. alba, primarily linalool, citral, and carvone, are known to have effects on the CNS. Among these activities, the anti-inflammatory, antiproliferative, antinociceptive, analgesic, anxiolytic, antidepressant, and neuroprotective have been documented [63, 64, 65, 66, 67, 68, 69]. Therefore, the findings mentioned above may explain the neuropharmacological therapeutic effects of L. alba observed in this bibliometric study” (lines 350-355).

Another concern raised was regarding limitations. The authors outlined the main limitations related to neuropharmacological activities and, in conclusion, presented future research directions for this plant investigation. These limitations include, among others: i) the lack of experimental models, as demonstrated in the sentences: “These findings suggest the potential use of L. alba as a phytotherapeutic agent for managing stress-related disorders, such as depression [77]. However, little is known about the sedative and anesthetic effects of the species in other experimental models, such as rodents” (lines 393-396); ii) side effects, as stated in the sentence: “However, toxicological effects were noted in one species of crab, which exhibited mortality of all subjects, but not sedative or anesthetic response [37]. Further studies are needed to assess the safety of L. alba as a sedative-anesthetic in different species” (lines 399-401); iii) and the scarcity of studies investigating the mechanisms of action underlying the neuropharmacological activities of Lippia alba, as indicated in “In rodents, an anxiolytic effect was also observed; however, the mechanisms of action were not investigated. This scenario suggests a broad area for future research on the anxiolytic and antidepressant effects of L. alba in rodents” (lines 430-433). Regarding its anticonvulsant activity, we included in the sentence “Few side effects were noted, particularly with regard to locomotor activity. The mechanisms of action of this activity have been minimally explored; however, GABAergic neurotransmission also appears to underlie the anticonvulsant properties of the plant [12, 53, 54]” (lines 437-439).

The authors also pointed out research gaps and future perspectives for L. alba studies, as follow: “Notably, the effects of the plant on other CNS disorders, such as depression and neurodegenerative diseases, remain unexplored, presenting a promising avenue for future research” (lines 462-464).

We appreciate the reference suggestion, which improved the revised manuscript.

  1. The novelty of this study must be emphasized in the abstract as well as conclusions.

Response: We appreciate the observation. We have revised the abstract “This bibliometric review provided new evidence reinforcing the potential of L. alba as a promising alternative for the treatment of neuropsychiatric disorders” (lines 34-39)—and the conclusion—“This unprecedent study represents a significant advance in the understanding of the neuropharmacological effects of Lippia alba, a plant widely distributed in the Americas, especially in the Amazon biome, and extensively used in traditional medicine. This species demonstrates the ability to interact and modulate specific mechanisms on the CNS, exhibiting sedative-anesthetic, anxiolytic, anticonvulsant, and analgesic activity, espe-cially through the GABAergic pathway as the main mechanism of action.” (lines 533-538) - to more clearly highlight the novelty of this study. We hope that the changes meet your recommendation.

  1. The neuroprotective mechanisms of L. alba must be highlighted in the entire manuscript.

Response: We appreciate your valuable observation. We have revised the manuscript to more clearly highlight the neuroprotective mechanisms of Lippia alba throughout the text. Additional information has been included, and existing points have been reinforced in the following sections:

  1. i) In abstract: “The main neuropharmacological activities identified were sedative-anesthetic, anxiolytic, anticonvulsant, and analgesic, with mechanisms of action via the GABAergic pathway” (lines 33-34).
  2. ii) In introduction: “In fact, preclinical studies demonstrate that the plant exhibits sedative-anesthetic actions, mainly in fish and aquatic invertebrates, by promoting a shorter anesthetic induction time and a longer anesthetic plane duration through the GABAergic pathway...” (lines 74-85).

iii) In discussion:

  1. a) For sedative-anesthetic activity: “The GABAergic pathway was identified as the primary mechanism of action. Additional mechanisms involved acetylcholinesterase (AChE) inhibition and reduced calcium influx [26, 41, 56] (lines 374-376)”, “In fish treated with L. alba, an inhibition of the HPI axis was observed, evidenced by a decrease in the levels of CRH, cortisol, and Hsp70 and Hsp90, along with an upregulation of POMCa [75, 76]. These findings suggest the potential use of L. alba as a phytotherapeutic agent for managing stress-related disorders, such as depression [77]. However, little is known about the sedative and anesthetic effects of the species in other experimental models, such as rodents (lines 392-397).”. Additionally, in the sentence: “Regarding the side effects of the sedative-anesthetic properties of L. alba, it is noteworthy that most studies have not reported changes in locomotor activity or cholinergic neurotransmission...” (lines 398-402).
  2. b) For anti-inflammatory and antioxidant activity: “In this context, it was investigated whether L. alba could influence neuroinflammation and oxidative stress. In fact, fish treated with L. alba essential oil showed an increase in the levels of key antioxidant enzymes (SOD, CAT, GPx, GSH, and Gr) and a decrease in lipid peroxidation in their brains, indicating a neuroprotective effect [30]. Additionally, there was inhibition of inflammatory pathways, as evidenced by the downregulation of nuclear factor kappa B (NF-κB) (lines 415-420)” “...This evidence suggests that L. alba may serve as a promising antioxidant and anti-inflammatory agent for the central nervous system (CNS). (lines 424-426).”
  3. c) For anxiolytic activity: “...These effects were attributed to GABAergic and serotonergic mechanisms. No side effects on locomotor activity or cortisol levels were noted at low doses [4, 31, 40]” (lines 429-431), demonstrating limitations in these studies: “In rodents, an anxiolytic effect was also observed; however, the mechanisms of action were not investigated” (lines 431-434).
  4. d) For anticonvulsant activity: “...These compounds have demonstrated sedative and anticonvulsant effects through the involvement of the GABAergic system and the antagonism of the glutamatergic system, respectively [86, 87]...” (lines 442-444).
  5. e) For dendritic arborization activity: “They observed that the dendritogenic action was mediated by the phosphatidylinositol 3-kinase (PI3Ks) pathway, which plays a crucial role in neuronal survival and dendrite growth” (lines 454-456).
  6. f) For analgesic activity and neuronal excitability inhibition: “Regarding preclinical studies, only one study investigated the mechanism of analgesic activity of L. alba, excluding the involvement of the opioidergic system [19]. L. alba also demonstrated inhibitory properties on nervous excitability and anticholinesterase activity [13, 16, 30]” (lines 459-461).

iiii) In conclusion: “The species exhibits sedative-anesthetic, anxiolytic, anticonvulsant, and analgesic activities, among others, with the GABAergic pathway being the main mechanism of L. alba” (lines 530-532).

We hope that these modifications meet your suggestion.

Reviewer 2 Report

Comments and Suggestions for Authors

This is an excellent and meticulously executed work by Silva et al. The bibliometric analysis is well-designed and clearly described. However, I feel that Tables 1 and 2 are overly lengthy. If the authors condense the information and present it more concisely, it would significantly enhance the article's readability and visibility.

Author Response

Dear Reviewer,

Firstly, we would like to thank you for your willingness to evaluate this review and for recognizing the scientific relevance of this paper. Below we have listed all your valuable considerations:

Reviewer #2

This is an excellent and meticulously executed work by Silva et al. The bibliometric analysis is well-designed and clearly described. However, I feel that Tables 1 and 2 are overly lengthy. If the authors condense the information and present it more concisely, it would significantly enhance the article's readability and visibility.

Dear reviewer,

We greatly appreciate the thoughtful suggestions provided for the manuscript, all of which have been carefully considered. Table 1 serves an essential purpose by presenting the articles selected for this review. To enhance its utility, we have added a brief summary of each selected article, ensuring that readers can gain a clear understanding of the theme and focus of each study.

As for Table 2, it plays a critical role in mapping the knowledge derived from the selected experimental studies. We acknowledge that this table is complex, and to address this, we prepared Figure 9, which summarizes the information from Table 2 in a more accessible and didactic format. In light of your meticulous feedback, we have decided to move Table 2 to the supplementary material, as we believe this adjustment will improve the overall flow and readability of the manuscript.

Reviewer 3 Report

Comments and Suggestions for Authors

Although the study looks interesting, there are major issues with this studies

A. Abstract:

1. The abstract mentions key aspects like “gaps in knowledge” but does not specifically state what these gaps are. It would be helpful to include one or two concrete examples.

2. Keywords such as "bibliometric analysis" and "essential oil" are effective, but consider adding terms like “phytochemistry” and “traditional medicine” for broader reach.

B. Introduction:

1. While the introduction effectively sets the stage, it could benefit from more explicit examples of previous successful discoveries from Lippia alba or similar plants to underline its importance.

C. Methods:

1. The criteria for article inclusion and exclusion (e.g., how disagreements between researchers were resolved) are noted but could benefit from more detail, such as whether any bias-checking mechanisms were implemented.

2. Although the methodological tools (e.g., VOSviewer, MapChart) are described, more information on how the software was used to ensure accuracy would be helpful.

D. Results and Discussion:

1. Overemphasis on Bibliometrics: The article places significant emphasis on bibliometric data. While this is informative, a deeper discussion of how the pharmacological findings can be translated into clinical or preclinical applications is needed.

1. Knowledge Gaps: The identified knowledge gaps should be elaborated upon. For instance, if certain neuropharmacological effects are understudied, specify these areas and their potential implications.

E. Comparative Analysis:

1. Comparing Lippia alba’s neuropharmacological effects with other plants of the Verbenaceae family might provide additional context.

F. Figures and Tables:

1. Figure 1: summarizing the methodological approach, is a good addition. However, visual representation of trends over time (e.g., number of publications by year) would enhance the reader’s understanding of the field's development.

2. A summary table of major neuropharmacological findings, with information on specific models, doses, and outcomes, would provide more clarity.

G. Citations:

1. The most cited studies are highlighted, but their significance to the broader field should be discussed more thoroughly. For instance, how have these studies influenced subsequent research?

H. Language and Style:

1. The manuscript generally reads well but contains minor grammatical and syntactical errors (e.g., “which conventional metrics were associated with a critical review…” in the abstract could be rephrased for clarity).

I. Discussion on Future Research Directions (Lines 357-375):

This section is strong but could expand on emerging CNS areas, such as Neuroinflammation and oxidative stress, as promising targets for L. alba.

J. Geographical Trends (Lines 294-303):

The focus on Brazil is appropriate but overshadows contributions from other countries. More analysis of how other regions could contribute would make this section more balanced.

K. Phytochemical Composition (Lines 359-369):

The variations in chemical composition are acknowledged but not analyzed in depth. A discussion on how environmental factors, extraction techniques, or chemotype variations influence pharmacological outcomes would add value.

L. Conclusion:

This article presents valuable insights into the neuropharmacological potential of Lippia alba and highlights critical knowledge gaps. By addressing methodological limitations, expanding on phytochemical and pharmacological findings, and emphasizing actionable future directions, the manuscript could significantly improve its impact and utility for researchers.

Comments on the Quality of English Language

The English could improve more clearly express the research

Author Response

Dear Reviewer,

We sincerely appreciate the detailed and systematized suggestions provided for our manuscript and have carefully considered each one. We would like to express our gratitude for your valuable time and effort in reviewing our work. Below we have listed all your valuable considerations:

Point A. Abstract:

  1. The abstract mentions key aspects like “gaps in knowledge” but does not specifically state what these gaps are. It would be helpful to include one or two concrete examples.

Response: We appreciate your observation. We have revised the abstract to include concrete examples of the knowledge gaps addressed in the study. Specifically, we have added the scarcity of studies comparing the various chemotypes of the species, the lack of understanding of the specific mechanisms of action of Lippia alba, and the limited research in emerging areas related to the central nervous system, such as mood and cognitive disorders, as follow: “The main neuropharmacological activities identified were sedative-anesthetic, anxiolytic, anti-convulsant, and analgesic, with mechanisms of action via the GABAergic pathway. This biblio-metric review provided new evidence reinforcing the potential of L. alba as a promising alternative for the treatment of neuropsychiatric disorders. It also highlighted existing knowledge gaps, mainly related to the comparison of the actions of the different chemotypes of the species and the investigation of the mechanisms underlying their neuropharmacological properties. Additionally, there is a lack of knowledge in other emerging areas related to the central nervous system, such as mood and cognitive disorders.” (lines 33–40). We hope these modifications make the abstract clearer and more informative.

  1. Keywords such as "bibliometric analysis" and "essential oil" are effective, but consider adding terms like “phytochemistry” and “traditional medicine” for broader reach.

  1. We appreciate your excellent comment on the keywords. They have been included to enhance the article's reach “Keywords: Lippia alba; Verbenaceae; medicinal plants; ethnopharmacology; essential oil; phy-tochemistry; traditional medicine; phytotherapy; bibliometric analysis; neuropharmacology” (lines 41–42).

Point B. Introduction:

  1. While the introduction effectively sets the stage, it could benefit from more explicit examples of previous successful discoveries from Lippia alba or similar plants to underline its importance

Response: Thank you very much for the suggestion. We have restructured the introduction to include clear examples of previous research that demonstrated the relevance of Lippia alba or similar plants, from both preclinical and clinical studies. These changes were made to reinforce the importance of the topic and provide a more solid foundation for the reader, as demonstrated in the sentence: “In fact, preclinical studies demonstrate that the plant exhibits sedative-anesthetic actions, mainly in fish and aquatic invertebrates, by promoting a shorter anesthetic induction time and a longer anesthetic plane duration through the GABAergic pathway. A neuroprotective effect has been associated with this property, particularly through the enhancement of enzymatic antioxidant capacity and the reduction of lipid peroxidation, as well as through anti-inflammatory action and a decrease in systemic stress, with few side effects. Other properties investigated include anxiolytic, anticonvulsant, analgesic, anticholinesterase, and inhibitory activities on CNS excitability [11, 12, 13]. Clinical studies indicate that L. alba serves as an alternative for the treatment of migraines in women, by reducing the frequency of episodes and the intensity of pain. These results suggest that L. alba exhibits promising neuropharmacological activities, highlighting several potential interactions with CNS targets [14]” (lines 74-85).

Point C. Methods:

  1. The criteria for article inclusion and exclusion (e.g., how disagreements between researchers were resolved) are noted but could benefit from more detail, such as whether any bias-checking mechanisms were implemented.

Response: We appreciate the suggestion, which we kindly accepted and incorporated in lines 106–113, in subsection 2.2. Study inclusion criteria and article selection of the materials and methods: "We included original and review articles investigating the neuropharmacological effects of Lippia alba, with no language restrictions. Two independent researchers followed a structured protocol: each abstract was thoroughly reviewed, and if the article approached neuropharmacological effects, it was selected for inclusion. Disagreements regarding article inclusion were resolved by a senior researcher, who conducted a full-text review and critical analysis to determine the final decision. Conference papers, editorials, and articles on Lippia alba that did not evaluate neuropharmacological properties were excluded, as detailed in the supplementary material” (linha 106 a 114).

  1. Although the methodological tools (e.g., VOSviewer, MapChart) are described, more information on how the software was used to ensure accuracy would be helpful.

Response: This is such an insightful suggestion. We have provided a more detailed description of the methodological tools applied in this review, as follows in lines 115-127, as follow: “For this study, we retrieved data from the WoS-CC database, including article titles, author names, citation counts, journal names, author keywords, countries, and institutions. Using VOSviewer software (version 1.6.16), we performed co-authorship analysis (based on publication and citation counts), evaluated keyword occurrences, and analyzed institutional contributions. The generated networks were interpreted as follows: each cluster represents an analysis item (e.g., authors, keywords, or institutions); the size of the cluster indicates the volume of publications or citations, the frequency of keyword occurrences, or institutional productivity; and the lines connecting clusters indicate co-authorship, keyword connections, or inter-institutional collaborations [17].

We also analyzed the relevance of journals, considering publication frequency and impact factor based on JCR 2023, ©2024 Clarivate. Additionally, we assessed the geographic distribution of selected articles using MapChart (https://mapchart.net/ accessed on August 12, 2024).”

Point D. Results and Discussion:

  1. Overemphasis on Bibliometrics: The article places significant emphasis on bibliometric data. While this is informative, a deeper discussion of how the pharmacological findings can be translated into clinical or preclinical applications is needed.

Response: We sincerely appreciate the reviewer's insightful comment. We recognize that while bibliometric analysis provides valuable insights into research trends and gaps, it should not overshadow the discussion of pharmacological findings and their translational potential. To address this issue, we revised the manuscript to provide greater emphasis on how the pharmacological discoveries related to Lippia alba can be translated into preclinical and clinical applications. Specifically, we have expanded the discussion on its neuroprotective mechanisms, potential therapeutic applications for neuropsychiatric disorders, and the challenges involved in transitioning from experimental models to clinical practice. These updates aim to provide a more balanced perspective, integrating bibliometric insights with a stronger focus on pharmacology and translational application. Additional information has been included, and existing points have been reinforced in the following sections:

  1. i) In abstract: “The main neuropharmacological activities identified were sedative-anesthetic, anxiolytic, anticonvulsant, and analgesic, with mechanisms of action via the GABAergic pathway” (lines 33-34).
  2. ii) In introduction: “In fact, preclinical studies demonstrate that the plant exhibits sedative-anesthetic actions, mainly in fish and aquatic invertebrates, by promoting a shorter anesthetic induction time and a longer anesthetic plane duration through the GABAergic pathway...” (lines 74-85).

iii) In discussion:

  1. a) For sedative-anesthetic activity: “The GABAergic pathway was identified as the primary mechanism of action. Additional mechanisms involved acetylcholinesterase (AChE) inhibition and reduced calcium influx [26, 41, 56]” (lines 374-376), as well as in “In fish treated with L. alba, an inhibition of the HPI axis was observed, evidenced by a decrease in the levels of CRH, cortisol, and Hsp70 and Hsp90, along with an upregulation of POMCa [75, 76]. These findings suggest the potential use of L. alba as a phytotherapeutic agent for managing stress-related disorders, such as depression [77]. However, little is known about the sedative and anesthetic effects of the species in other experimental models, such as rodents” (lines 392-397). Additionally, in the sentence: “Regarding the side effects of the sedative-anesthetic properties of L. alba, it is noteworthy that most studies have not reported changes in locomotor activity or cholinergic neurotransmission...” (lines 398-402).
  2. b) For anti-inflammatory and antioxidant activity: “In this context, it was investigated whether L. alba could influence neuroinflammation and oxidative stress. In fact, fish treated with L. alba essential oil showed an increase in the levels of key antioxidant enzymes (SOD, CAT, GPx, GSH, and Gr) and a decrease in lipid peroxidation in their brains, indicating a neuroprotective effect [30]. Additionally, there was inhibition of inflammatory pathways, as evidenced by the downregulation of nuclear factor kappa B (NF-κB)” (lines 415-420), as well as in: “...This evidence suggests that L. alba may serve as a promising antioxidant and anti-inflammatory agent for the central nervous system (CNS).” (lines 424-426).
  3. c) For anxiolytic activity: “...These effects were attributed to GABAergic and serotonergic mechanisms. No side effects on locomotor activity or cortisol levels were noted at low doses [4, 31, 40]” (lines 429-431), demonstrating limitations in these studies: “In rodents, an anxiolytic effect was also observed; however, the mechanisms of action were not investigated” (lines 431-434).
  4. d) For anticonvulsant activity: “...These compounds have demonstrated sedative and anticonvulsant effects through the involvement of the GABAergic system and the antagonism of the glutamatergic system, respectively [86, 87]...” (lines 442-444).
  5. e) For dendritic arborization activity: “They observed that the dendritogenic action was mediated by the phosphatidylinositol 3-kinase (PI3Ks) pathway, which plays a crucial role in neuronal survival and dendrite growth” (lines 454-456).
  6. f) For analgesic activity and neuronal excitability inhibition: “Regarding preclinical studies, only one study investigated the mechanism of analgesic activity of L. alba, excluding the involvement of the opioidergic system [19]. L. alba also demonstrated inhibitory properties on nervous excitability and anticholinesterase activity [13, 16, 30]” (lines 459-461).

iiii) In conclusion: “This species demonstrates the ability to interact and modulate specific mechanisms on the CNS, exhibiting sedative-anesthetic, anxiolytic, anticonvulsant, and analgesic activity, especially through the GABAergic pathway as the main mechanism of action.” (lines 530-532).

We hope that these modifications meet your suggestion.

  1. Knowledge Gaps: The identified knowledge gaps should be elaborated upon. For instance, if certain neuropharmacological effects are understudied, specify these areas and their potential implications.

Response: We sincerely appreciate the reviewer's insightful comment.  The authors outlined the main gaps in research regarding the neuropharmacological activities of Lippia alba and, in conclusion, presented future directions for studies on the plant. These limitations include, among others:

  1. i) The limited use of experimental models, as demonstrated in the sentence: “These findings suggest the potential use of L. alba as a phytotherapeutic agent for managing stress-related disorders, such as depression [77]. However, little is known about the sedative and anesthetic effects of the species in other experimental models, such as rodents” (lines 393-396);
  2. ii) Side effects, as stated in the sentence: “However, toxicological effects were noted in a species of crab, which exhibited mortality of all subjects, but not sedative or anesthetic response [37]. Further studies are needed to as-sess the safety of L. alba as a sedative-anesthetic in different species” (lines 399-401).

iii) Most importantly, the scarcity of studies investigating the mechanisms of action underlying the neuropharmacological activities of Lippia alba, as indicated in: “In rodents, an anxiolytic effect was also observed; however, the mechanisms of action were not investigated. This scenario suggests a broad area for future research on the anxiolytic and antidepressant effects of L. alba in rodents” (lines 430-433).

Or for its anticonvulsant activity, as demonstrated in: “Few side effects were noted, particularly with regard to locomotor activity. The mechanisms of action of this activity have been minimally explored; however, GABAergic neurotransmission also appears to underlie the anticonvulsant properties of the plant [12, 53, 54]” (lines 437-439).

The authors also pointed out future research perspectives for the species, as indicated in the sentence: “Notably, the effects of the plant on other CNS disorders, such as depression and neurodegenerative diseases, remain unexplored, presenting a promising avenue for future research” (lines 462-464).

Point E. Comparative Analysis:

  1. Comparing Lippia alba’s neuropharmacological effects with other plants of the Verbenaceae family might provide additional context.

Response: We appreciate the valuable suggestion. We have incorporated into the discussion section a comparison of the neuropharmacological effects of Lippia alba with other plants from the Verbenaceae family. These comparisons were added to provide additional context and highlight both the similarities and the unique characteristics of Lippia alba, as follow: “Based on all this evidence, Lippia alba stands out among other plants of the Lippia genus in the Verbenaceae family for presenting the greatest range of studied neuropharmacological activities and mechanisms of action. Studies involving species such as Lippia multiflora, Lippia gracilis, Lippia grata, Lippia origanoides, Lippia graveolens, Lippia geminata, and Lippia adoensis demonstrate that these species exhibit prominent analgesic activity, with few studies investigating other neuropharmacological activities for this genus of plants [96]” (lines 480-485). We hope these modifications meet your recommendation.

 Point F. Figures and Tables:

  1. Figure 1: summarizing the methodological approach, is a good addition. However, visual representation of trends over time (e.g., number of publications by year) would enhance the reader’s understanding of the field's development.

Response: What an assertive proposal! Adding a figure to explore the number of publications by year would indeed enhance our manuscript. Consequently, we have included a newsection in the Results titled “Scientific Production by Decades”, as demonstrated in the sentence: “A clear majority of publications occurred in the 2010s (n=34), significantly more than in the 2020s (n=12), 2000s (n=4), and 1990s (n=2). This decade also received the most citations (845), exceeding the 2000s (192), 2020s (144), and 1990s (106). See Figure 7 for a comprehensive overview.” (Linhas 213-217). The figure 7 illustrated the publication trends over the decades.

  1. A summary table of major neuropharmacological findings, with information on specific models, doses, and outcomes, would provide more clarity.

Response: This point was somewhat unclear to us, as we have already provided a detailed table of the science mapping (now included in the supplementary material, Table 3) and as schematic figure summarizing the main neuropharmacological outcomes (Figure 9, page 22). We kindly invite you to apreciate these materials, which we believe address your suggestions.

Point G. Citations

  1. The most cited studies are highlighted, but their significance to the broader field should be discussed more thoroughly. For instance, how have these studies influenced subsequent research?

Response: We apologize for this oversight. In fact, we did not provide a discussion on the relevance of the most cited papers. Thanks to your meticulous observations, we have added this information, as follows in lines 296-306, as demonstrated in the sentence: “Regarding publications by decade, the 2010s were by far the most productive decade, with 34 publications and the highest number of citations (945). Although still ongoing, the 2020s demonstrate increasing research activity, with 12 articles published to date and a growing citation count (144). Notably, despite having fewer publications (4 in the 2000s and 2 in the 1990s) compared to more recent decades, these earlier periods still garnered substantial citation numbers (192 and 106, respectively), highlighting the enduring relevance of early work. These metrics, in addition to the number of studies, reflect sustained interest in the topic. Another relevant parameter to analyze is the type of studies most published. Among the 52 articles selected for this review, in vivo preclinical studies were the most prevalent (n = 38), followed by reviews (n = 8), in vitro studies (n = 3), clinical studies (n = 2) and studies using combined in vitro and in silico models (n = 1)”.

Point H. Language and Style:

  1. The manuscript generally reads well but contains minor grammatical and syntactical errors (e.g., “which conventional metrics were associated with a critical review…” in the abstract could be rephrased for clarity).

Response: We really apologize for these typos. We extensively revised the English grammar to improve the revised manuscript.

Point I. Discussion on Future Research Directions (Lines 357-375):

This section is strong but could expand on emerging CNS areas, such as Neuroinflammation and oxidative stress, as promising targets for L. alba.

Response: We appreciate the excellent recommendation. We have expanded the discussion section to include neuroinflammation and oxidative stress as emerging areas and promising targets for the effects of Lippia alba, as demonstrated in the sentences: “In this context, it was investigated whether L. alba could influence neuroinflammation and oxidative stress. In fact, fish treated with L. alba essential oil showed an increase in the levels of key antioxidant enzymes (SOD, CAT, GPx, GSH, and Gr) and a decrease in lipid peroxidation in their brains, indicating a neuroprotective effect [30]. Additionally, there was inhibition of inflammatory pathways, as evidenced by the downregulation of nuclear factor kappa B (NF-κB) [30]” (lines 415-420).

Furthermore, we provided evidence of how antioxidant and anti-inflammatory agents can be promising therapies for the treatment of central nervous system diseases: “Oxidative stress and neuroinflammation play central roles in the development and progression of neuropsychiatric disorders, including depression, schizophrenia, bipolar disorder, Alzheimer's, and Parkinson's [78, 79, 80, 81]. This is primarily due to the brain's high oxygen consumption, which induces vulnerability to oxidative stress compared to other organs [82].” (lines 403-406).

These new sentences enrich the section and reinforce its relevance for the development of new therapeutic approaches for the CNS: “This evidence suggests that L. alba may serve as a promising antioxidant and anti-inflammatory agent for the central nervous system (CNS)” (lines 424-426). The authors also discuss the importance of using and studying L. alba in other aspects of the CNS, as demonstrated in the sentence: “Notably, the effects of the plant on other CNS disorders, such as depression and neurodegenerative diseases, remain unexplored, presenting a promising avenue for future research” (lines 463-465).

Point J. Geographical Trends (Lines 294-303):

The focus on Brazil is appropriate but overshadows contributions from other countries. More analysis of how other regions could contribute would make this section more balanced.

Response: We appreciate the comment. We have included analyses of studies conducted in other regions, balancing the discussion and broadening the perspective beyond the focus on Brazil. We hope these additions meet your suggestion: “Considering geographic criteria, global publications on Lippia alba and its neuropharmacological properties reveal a significant concentration of scientific output in Brazil, which holds the highest number of publications and citations. This leadership is likely attributed to the species' origin and the region's rich biodiversity. Following Brazil, the United States, Spain, and Colombia each contributed two publications. Despite having fewer publications, the United States and Spain achieved notable citation counts, with 71 and 40 citations, respectively, while Colombia received 5 citations. Other countries produced only one publication each. These findings underscore Brazil's leading role in this area of research. However, it is important to emphasize that other countries have also made significant contributions to understanding the neu-ropharmacological properties of the species over the years. For instance, Iran, Turkey, Spain, Japan, and France have made key contributions to understanding its sedative-anesthetic action (8, 24, 25, 26, 27). Additionally, Guatemala and the United States have provided excellent reviews on the plant and its biological activities (10, 28). Colombia has presented a study on the dendritogenic potential and a detailed review of L. alba (16, 29). These data highlight the fundamental contributions of these countries to advancing the understanding of the neuropharmacological properties of Lippia alba” (Lines 307–323).

Point K. Phytochemical Composition (Lines 359-369):

The variations in chemical composition are acknowledged but not analyzed in depth. A discussion on how environmental factors, extraction techniques, or chemotype variations influence pharmacological outcomes would add value.

Response: We appreciate the reviewer's insightful comment. We recognize the importance of a more in-depth discussion on the chemical composition variations of Lippia alba and their impact on pharmacological outcomes. To address this issue, we have expanded our discussion to consider how environmental factors, extraction techniques, and chemotypic variations influence the plant’s bioactive profile and, consequently, its pharmacological effects. We hope these modifications enhance the discussion in the manuscript and align with the journal’s expectations, as the sentence demonstrates: “Chemical composition is essential in medicinal plant research, as it validates pharmacological findings and helps elucidate underlying mechanisms of action [6, 24, 29]. Notably, a significant proportion of the reviewed studies lacked detailed phytochemical analyses. Studies reporting the phytochemical composition revealed varied primary constituents of L. alba in different neuropharmacological activities. For example, linalool and citral predominated in sedative-anesthetic studies, while linalool, carvone, citral, and limonene were more common in anxiolytic and anticonvulsant studies. Analgesic studies frequently identified citral, geraniol, and carvone. Furthermore, pharmacological effects may arise from a synergistic combination of compounds, termed a phytocomplex, rather than from a single constituent [19, 34, 44, 54] (lines 487-493)” ; “In fact, several studies have indicated that not only species of the Lippia genus, but also various plants of the Verbenaceae family exhibit wide variation in the components of their essential oils, displaying a broad range of chemotypes for these species [61]. Several factors can influence the chemotype of the species, including the region of the plant used, the harvest time, the collection location, seasonal variations, climate, extraction method, and abiotic factors [62]” (lines 345-350).

We thank again the Editor and the Reviewers for their comments and helpful suggestions, which contributed to allow us improving the MS. We hope that, by addressing all the comments of the Reviewers, the revised version of the MS may prove acceptable for publication in  Pharmaceuticals.

Sincerely yours,

Cristiane Maia

Round 2

Reviewer 1 Report

Comments and Suggestions for Authors

Authors have addressed all the issues.